# Open fire exposure increases the risk of pregnancy loss in South Asia

Tao Xue [1,5,6 ✉], Guannan Geng [2,5], Yiqun Han[3], Huiyu Wang[1], Jiajianghui Li[1], Hong-tian Li[1], Yubo Zhou[1] & Tong Zhu[4,6]

Interactions between climate change and anthropogenic activities result in increasing numbers of open fires, which have been shown to harm maternal health. However, few studies have examined the association between open fire and pregnancy loss. We conduct a self-comparison case-control study including 24,876 mothers from South Asia, the region with the heaviest pregnancy-loss burden in the world. Exposure is assessed using a chemical transport model as the concentrations of fire-sourced $PM_{2.5}$ (i.e., fire $PM_{2.5}$). The adjusted odds ratio (OR) of pregnancy loss for a 1-$\mu g/m^3$ increment in averaged concentration of fire $PM_{2.5}$ during pregnancy is estimated as 1.051 (95% confidence intervals [CI]: 1.035, 1.067). Because fire $PM_{2.5}$ is more strongly linked with pregnancy loss than non-fire $PM_{2.5}$ (OR: 1.014; 95% CI: 1.011, 1.016), it contributes to a non-neglectable fraction (13%) of $PM_{2.5}$-associated pregnancy loss. Here, we show maternal health is threaten by gestational exposure to fire smoke in South Asia.

[1] Institute of Reproductive and Child Health / Ministry of Health Key Laboratory of Reproductive Health and Department of Epidemiology and Biostatistics, School of Public Health, Peking University Health Science Centre, Beijing, China. [2] School of Environment, Tsinghua University, Beijing, China. [3] Environmental Research Group, MRC Centre for Environment and Health, Imperial College London, London, UK. [4] BIC-ESAT and SKL-ESPC, College of Environmental Science and Engineering, Peking University Beijing, Beijing, China. [5] These authors contributed equally: Tao Xue, Guannan Geng. [6] These authors jointly supervised: Tao Xue, Tong Zhu. ✉email: xuetaogk_9032@126.com

In recent decades, open fire events have been increasingly reported around the world, raising significant public awareness of the adverse socioeconomic and health impacts of open fires[1]. Open fires comprise several different types including wildfires, mountain fires, coal mining fires, and slash-and-burn agriculture, and the sources can be directly related to human activities[2] or indirectly related via climate change[3]. Open fires have been reported to harm human health primarily by increasing ambient exposure to hazardous chemicals[4]. Biomass burning emits massive toxic air pollutants such as particulate matter (PM), polycyclic aromatic hydrocarbons, and volatile organic compounds[5]. Global estimates indicate that 339,000 premature deaths per year can be attributed to exposure to open fire smoke[6] through increased risk of cardiorespiratory diseases including cardiac arrest, asthma, hypertension, and respiratory infections[4,5]. Due to global warming, the number of such events is expected to increase as extreme heat events become more frequent[1]. Understanding the mechanisms underlying the health impacts of open fires is critical to preventing the related disease burden.

Pregnant women and their embryos can be more susceptible to environmental hazards than the general adult population. According to recent epidemiological studies of preterm birth and low birthweight infants[7,8], gestational exposure to open fires can restrict fetal growth, which, in severe circumstances, can increase prematurely terminated gestation[9], also known as pregnancy loss (miscarriage or stillbirth). Thus, the association between pregnancy loss and open fire exposure is biologically plausible. Additionally, similarities between open fire smoke and ambient fine particles ($PM_{2.5}$) in terms of exposure patterns and chemical species suggest that they likely share some common health outcomes. $PM_{2.5}$ exposure has been associated with increased risk of pregnancy loss[10–12], as is gestational exposure to open fire smoke, at least theoretically. Because PM from biomass burning is richer in toxic organic components than typical ambient PM in the environment[13], the effect of open fires on pregnancy may be greater than that of $PM_{2.5}$ and thus should be further examined.

Pregnancy loss is an insufficiently studied disease burden for many reasons, such as the associated stigma and lack of awareness of its adverse health impacts[14]. Pregnancy loss not only directly harms the mother's physical health (e.g., increasing the risk of infertility) but also adversely affects the whole family through psychosocial and socioeconomic pathways[15]. Additionally, the geographic distribution of the burden of pregnancy loss is unequal. Due to high rates of low- or middle-income levels and high fertility rates, low-latitude countries (e.g., countries in South Asia and Africa), which are also hotspots for open fires due to climate characteristics, have the highest baseline risk of pregnancy loss[16]. Given this pattern, examining the epidemiological link between open fire exposure and pregnancy loss is of public health importance.

To test the hypothesis that open fire exposure increases the risk of pregnancy loss, we conducted an epidemiological study in three South Asian countries, India, Pakistan, and Bangladesh, which have the highest rates of pregnancy loss in the world[16], together contributing approximately one-third (35.3%) of all stillbirths in the world despite having only one-quarter (25.8%) of all newborns. Exposure was assessed using three alternative indicators of open fire: (1) satellite imaging data, (2) emissions data, and (3) concentrations of fire $PM_{2.5}$, estimated from chemical transport modeling (CTM) outputs. Due to good interpretability, the CTM-based indicator was used as the main metric in our epidemiological analyses. To examine the association of these exposure indicators with pregnancy loss, we further applied a well-derived self-comparison case–control design that has been used in our previous work on ambient $PM_{2.5}$[10].

## Results

**Descriptive statistics.** This study examined 24,876 cases of pregnancy loss and 50,386 matched controls (normal deliveries) during 2000–2014. Of these, 11.5% were in Bangladesh, 12.7% in Pakistan, and the rest were in India. The mean age at pregnancy loss was 26.15 years (standard deviation [SD] = 5.76 years), which was older than that at normal delivery (mean = 24.48 years; SD = 5.06 years). More details of the study population are presented in Supplementary Table 1. The spatial distribution of the studied samples is visualized in Fig. 1.

For all samples, the mean total $PM_{2.5}$ was 53.2 μg/m³, of which open fires contributed 1.2 μg/m³. However, the fire $PM_{2.5}$ distribution displayed an extremely right-tailed bias. The long-term mean at hotspot locations (Fig. 1a) or areal average during the peak seasons (Fig. 1b) would be approximately 5 μg/m³. In particular, the spatiotemporal pattern of fire $PM_{2.5}$ clearly differed from that of total $PM_{2.5}$ (Fig. 1b and Supplementary Fig. 1). For instance, the forest area of Northeast India, where the total concentration of $PM_{2.5}$ was low (Supplementary Fig. 1), suffered most from open fires due to cultivation practices[17,18]. The divergence in spatiotemporal distribution made the open fires a considerable contributor to local $PM_{2.5}$ exposure.

Among all cases of pregnancy loss, the average gestational exposure to fire $PM_{2.5}$ was 1.30 μg/m³, which was greater than that among controls (1.14 μg/m³). The long-term mean area of satellite data or GFED dry-matter emissions showed similar results (Supplementary Table 1). All three indicators consistently suggested that pregnancy loss was positively correlated with high levels of fire exposure.

**Associations by different exposure indicators.** Table 1 presents the associations between fire $PM_{2.5}$ and pregnancy loss, as estimated by different regression models. According to the fully adjusted model, each increment of 1 μg/m³ $PM_{2.5}$ was associated with a 5.1% (95% CI: 3.5%, 6.7%) increase in the risk of pregnancy loss. Additionally, the direction of the estimated association between fire and pregnancy loss was not changed by using different indicators of exposure. Thus, all three indicators consistently suggested adverse effects of fire $PM_{2.5}$ on pregnancy loss. However, the significance level of the ORs estimated by satellite data on burned area or dry-matter emissions data was sensitive to adjustments of different covariates. A possible explanation is the potential for misclassification of exposure (due to multiple reasons, e.g., spatial resolution or accuracy) when using satellite or emissions data.

**Sensitivity analyses for the estimated associations.** We first examined whether the estimated association between fire $PM_{2.5}$ and pregnancy loss differed among subpopulations (Fig. 2). We found that the estimated association did not vary significantly among subgroups for most population characteristics (e.g., educational level, residence, or insurance coverage), but did differ with maternal age. We found that pregnancy at an older age was more susceptible to the adverse effects of fire $PM_{2.5}$ on the woman. For mothers with a maternal age <30, 30–34, or ≥35 years, each increment of 1 μg/m³ $PM_{2.5}$ was associated with an excess pregnancy-loss risk of 4.1% (95% CI: 2.4%, 5.8%), 7.6% (95% CI: 4.0%, 11.3%), or 11.1% (95% CI: 5.5%, 17.0%), respectively.

Next, we ran further sensitivity analyses focusing on specific subgroups of the data (Fig. 3). Figure 3 shows that the direction of the association between fire $PM_{2.5}$ and pregnancy loss was not changed when different inclusion criteria were applied to the samples. Except for a small subgroup (i.e., stillbirths and their controls), all of the estimated associations

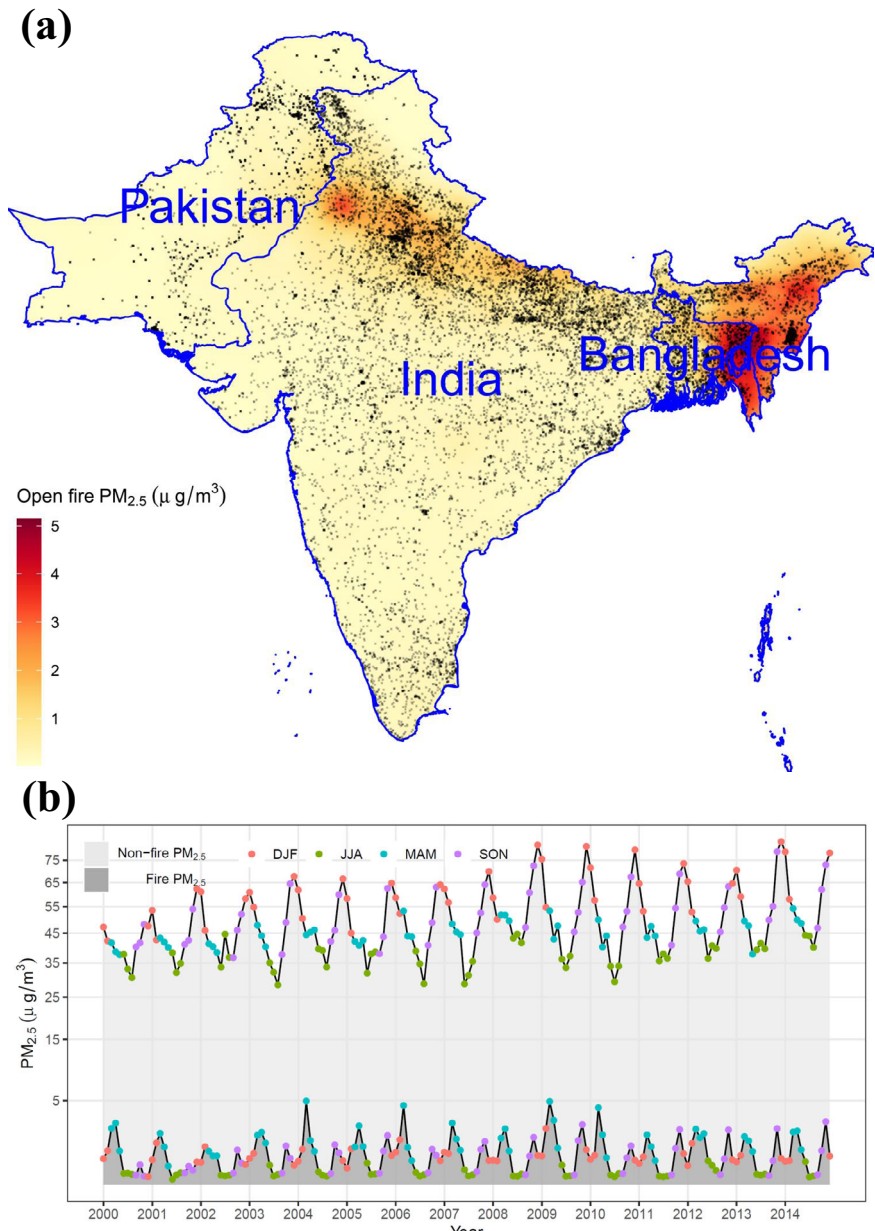

**Fig. 1 Spatial and temporal variations in the concentrations of PM$_{2.5}$ attributable to open fire.** Panel **a** presents the mean concentration of open fire PM$_{2.5}$ during 2000–2014 with locations of surveyed samples (black dots). Panel **b** presents the population-weighted means of open fire or non-fire PM$_{2.5}$ by month. The map in Panel **a** is generated by the corresponding author (T.X.) using the data from Natural Earth (https://www.naturalearthdata.com/). PM$_{2.5}$: particulate matter with a diameter of less than 2.5 μm; DJF: December–January–February; MAM: March–April–May; JJA: June–July–August; SON: September–October–November.

**Table 1 Estimated associations between pregnancy loss and three indicators of open fire exposure.**

| Adjusted for covariates* | PM$_{2.5}$ mutually adjusted[†,‡] | Odds ratio per increment | | |
|---|---|---|---|---|
| | | 1 μg/m$^3$ fire PM$_{2.5}$[†] | 1% satellite burned area | 10 g/m$^3$/month dry-matter emissions |
| No | No | 1.064 (1.051, 1.076) | 1.013 (1.003, 1.024) | 1.064 (1.030, 1.099) |
| | Yes | 1.039 (1.027, 1.051) | 1.015 (1.005, 1.025) | 1.056 (1.023, 1.089) |
| Yes | No | 1.068 (1.052, 1.084) | 1.011 (0.999, 1.024) | 1.044 (1.003, 1.087) |
| | Yes | 1.051 (1.035, 1.067) | 1.013 (1.000, 1.025) | 1.037 (0.996, 1.079) |

* The adjusted covariates included maternal age, temperature, humidity, and temporal trends.
[†] Fire PM$_{2.5}$ and non-fire PM$_{2.5}$ were estimated from CTM simulations.
[‡] Fire PM$_{2.5}$ and non-fire PM$_{2.5}$ were simultaneously incorporated into the regression model.

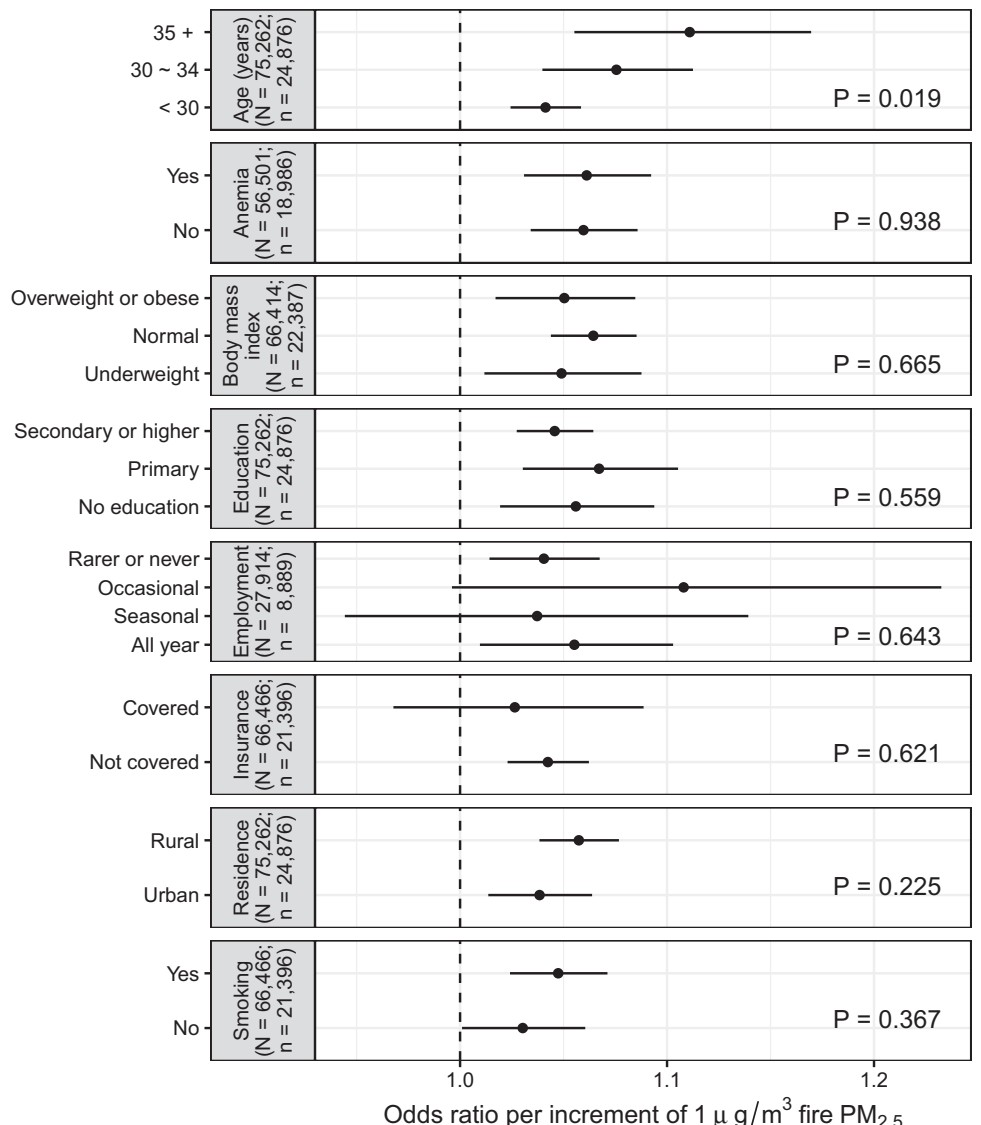

**Fig. 2 Estimated associations between open fire PM$_{2.5}$ and pregnancy loss by subpopulation.** Concentrations of fire PM$_{2.5}$ were assessed by the GEOS-Chem model, and the associations were adjusted by maternal age, temperature, humidity, temporal trends, and non-fire PM$_{2.5}$. Along y-axis, "n" denotes the number of mothers, and "N" denotes the size of total samples (i.e., cases and controls). In each panel, "P" presents the likelihood-ratio Chi-squared test P-value for an alternative hypothesis that the estimated associations are different between subgroups classified by a subpopulation indicator. Each group of estimates is derived from an individual model of "n" independent pairs of cases and controls. The dots denote point-estimates of the associations, and the error bars denote the corresponding 95% confidence intervals. PM$_{2.5}$ particulate matter with a diameter of less than 2.5 μm.

were significantly positive. The results further enhanced the robustness of our finding on the adverse effects of fire PM$_{2.5}$. For instance, restricting the analyzed samples by different recall periods did not change the significance level of the estimated associations. Here, it is worth highlighting the results from two subgroup analyses: (1) cases and their bidirectional controls, and (2) cases and controls exposed to transported fire PM$_{2.5}$. The former analysis applied a strict control for the confounding effects of temporal trends, and the latter could potentially preclude non-air-pollution channels (e.g., destroying settlements) to explain the association between fire and pregnancy loss.

Finally, we estimated the nonlinear association between fire PM$_{2.5}$ and pregnancy loss (Fig. 4). To examine the hypothesis of linear association, we used a likelihood ratio test, which compared the nonlinear model with the fully adjusted linear model. The test result indicated a nonlinear association (P-value <0.001). Figure 4 showed a super-linear exposure–response function for the effects of PM$_{2.5}$,

which indicated a higher marginal risk for a larger fire. We also ran a model with the categorical variable of fire PM$_{2.5}$ (Supplementary Fig. 2), which further confirmed the super-linear relationship.

**Comparative impacts from fire and non-fire PM$_{2.5}$.** Figure 5 presents the comparison between the impact of fire PM$_{2.5}$ and that of non-fire PM$_{2.5}$. The public health importance of fire PM$_{2.5}$ can differ from that of non-fire PM$_{2.5}$ in terms of effect magnitudes, exposure levels, and spatial distributions. Although the exposure to high concentrations of fire PM$_{2.5}$ can be less common than that to non-fire PM$_{2.5}$ (Fig. 1), we found that the former had a stronger association. According to the adjusted model, a 1-μg/m$^3$ increment in the non-fire PM$_{2.5}$ was associated with a 1.4% (1.1%, 1.6%) increase in pregnancy loss. The association was lower than that for fire PM$_{2.5}$ (5.1%; 95% CI: 3.5–6.7%, Table 1). Given this observation, in our study domain, fire PM$_{2.5}$, which only accounted for 2% of total PM$_{2.5}$, nonetheless contributed to 13%

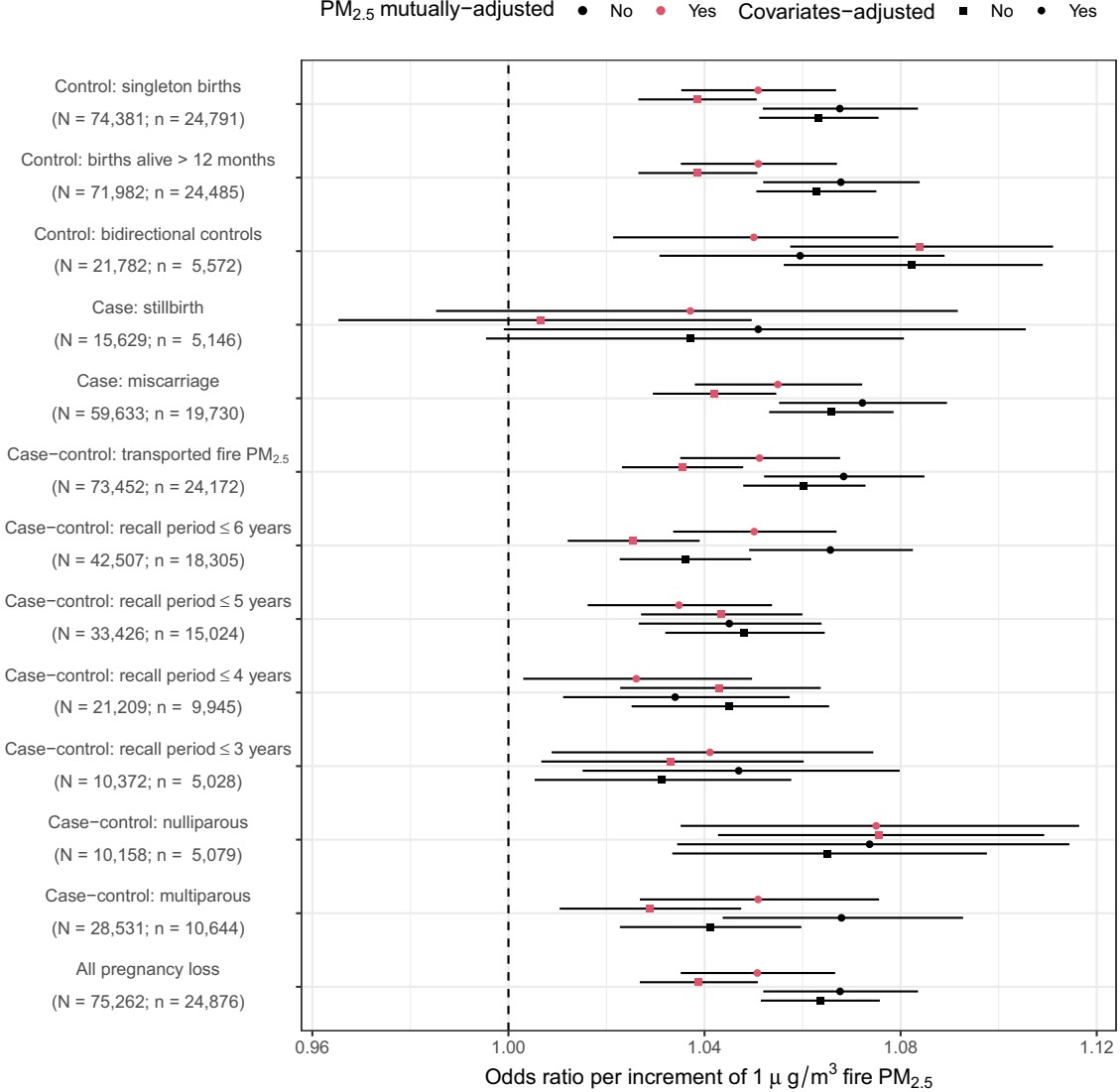

**Fig. 3 Estimated associations between open fire PM$_{2.5}$ and pregnancy loss for the selected subsets.** Concentrations of fire PM$_{2.5}$ were assessed by the GEOS-Chem model, and the associations were adjusted by maternal age, temperature, humidity, temporal trends, and non-fire PM$_{2.5}$. Along y-axis, "n" denotes the number of mothers, and "N" denotes the size of total samples (i.e., cases and controls). Each estimate is derived from an individual model of "n" independent pairs of cases and controls. The dots denote point-estimates of the associations, and the error bars denote the corresponding 95% confidence intervals. PM$_{2.5}$ particulate matter with a diameter of less than 2.5 μm.

of PM$_{2.5}$-linked pregnancy loss. Additionally, in different countries, the importance of fire PM$_{2.5}$ varied. The attributable fraction of fire PM$_{2.5}$ was highest in Bangladesh, followed by that in India and Pakistan (Fig. 5).

## Discussion

This study identified a significant association between exposure to open fire and increased risk of pregnancy loss in India, Pakistan, and Bangladesh. The association was moderately robust to different model settings in terms of exposure indicator, covariate adjustment, subpopulation stratification, and a few selection criteria of controls. This is the first study to report the epidemiological linkage between open fire and pregnancy loss at a subcontinental scale.

Although research on the association between fire smoke and pregnancy loss is limited, reports of the adverse effects on clinical outcomes of fetal growth retardation are common. Holstius et al. found that maternal exposure to the 2003 Southern California Wildfires was associated with lower birthweights[7]. Abdo et al.

found that exposure to wildfire smoke PM$_{2.5}$ was associated with preterm birth and decreased birthweight in Colorado[8]. Additionally, occupational exposure to fire was associated with increased risk of miscarriage and preterm birth[19]. In previous studies on the health impacts of open fire, exposure to airborne particles from fire smoke was considered the major exposure pathway[4]. Thus, studies that reported an association between pregnancy loss and PM$_{2.5}$ from other sources can be regarded as indirect evidence, supporting our findings. In our previous study in Africa[10], we reported a robust association between PM$_{2.5}$ and pregnancy loss and found that the association was potentially increased by the biomass burning components of PM$_{2.5}$ (e.g., black carbon). A recent study in London reported that exposure to traffic-related PM$_{2.5}$ increased the risk of stillbirth[20]. A meta-analysis reported a pooled effect estimate of a 2.1% (−0.4%, 4.6%) increase in stillbirth per 4-μg/m$^3$ increment in PM$_{2.5}$[12]. In our study, pregnancy loss was simultaneously associated with non-fire PM$_{2.5}$ and fire PM$_{2.5}$, in agreement with the extant findings.

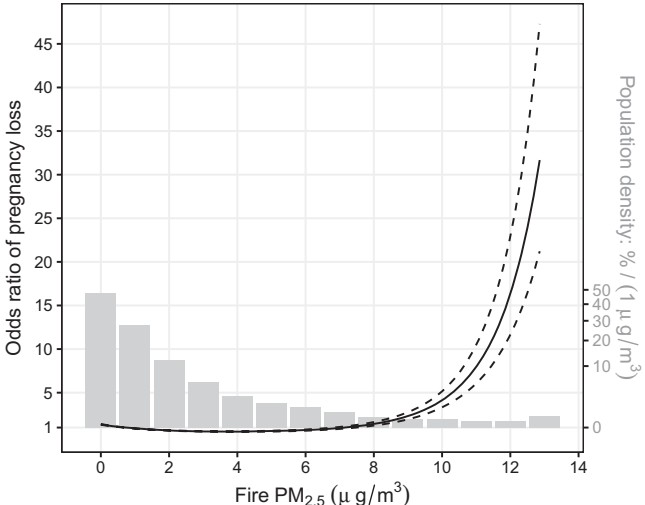

**Fig. 4 Estimated nonlinear associations between open fire PM$_{2.5}$ and pregnancy loss.** Concentrations of fire PM$_{2.5}$ were assessed by the GEOS-Chem model, and the associations were adjusted by maternal age, temperature, humidity, temporal trends, and non-fire PM$_{2.5}$. The estimates are derived from the full-adjusted model with 24,876 independent mothers. The solid line denotes point-estimates of the nonlinear association, and the dashed lines denote the corresponding point-wise 95% confidence intervals. PM$_{2.5}$ particulate matter with a diameter of less than 2.5 μm.

We also found a super-linear exposure–response between fire PM$_{2.5}$ and pregnancy loss (Fig. 4). The result suggests that the health impacts from large open fires, such as forest fires, should be considered. Our analysis of type-specific associations between fire emissions and pregnancy loss showed a consistent result (Supplementary Fig. 3). Stronger associations were found for the large fires (i.e., grassland, shrubland fires, and temperate forest fires) compared to the small fires (i.e., deforestation and degradation). However, the interpretation should be cautious for two reasons: first, by quantifying the exposure using fire PM$_{2.5}$ concentrations, our analyses ignored the complex behavior of individual fires, which can be different in terms of size, duration, and speed[21]. All dimensions of fire behavior can affect the exposure pattern and chemical species of fire smoke, and thus the relevant health effect. Second, since frequency for large fires is low, our estimates at the high exposure levels are with low confidence due to the limited sample size (Fig. 4 and Supplementary Fig. 3). Therefore, future studies are needed to confirm or refute our findings on the health effect of large fires.

Driven by interactions between climate change and anthropogenic activities, frequent wildfires can dramatically affect many aspects of human sustainability[22], including air quality, ecological diversity, distribution of infectious disease vectors, and public health. Previous studies exploring the health effects of open fires focused on respiratory and cardiovascular diseases[5] but overlooked the impacts on susceptible individuals, such as infants and pregnant women. Unlike urban PM$_{2.5}$ pollution, which is prolonged, open fires occur only occasionally, but most lead to extremely high levels of exposure. Some subclinical negative outcomes (e.g., blood pressure elevation), which can be reversed after blocking the risk factor, are threatened by prolonged exposures (e.g., urban PM$_{2.5}$ pollution). Compared to those outcomes, the irreversible ones (e.g., adverse birth outcomes) are more threatened by occasionally peak exposures (e.g., fire PM$_{2.5}$). Among various adverse birth outcomes (e.g., preterm birth, and low birthweight) that have been associated to PM$_{2.5}$ exposure[7,8], pregnancy loss may be the most dangerous because it not only reflects the most severe damage to the fetus but also will increase

the risk of other outcomes in subsequent deliveries (e.g., subsequent preterm birth)[23]. Therefore, pregnancy loss can be a tool to examine key impacts from open fires, and the relevant findings from this study reveal the importance of such exposure in terms of health outcomes. We found that the excess risk of fire PM$_{2.5}$ was 269% (149%, 411%) higher than that of non-fire PM$_{2.5}$ (Fig. 5). Because of the high OR for per-unit exposure[13], open fires contributed only a small amount to the concentration of PM$_{2.5}$ (2%) but a non-negligible fraction of the PM$_{2.5}$-associated disease burden (13%). Therefore, priority should be given to the prevention of maternal exposure to open fire smoke.

Studying the health effects of open fire PM$_{2.5}$ not only has public health significance but can also be useful to reveal the causal relationship between air quality and disease. In most places in the world, air pollutants co-vary greatly with anthropogenic emissions, and their health effects can thus be confounded by socioeconomic factors. Decoupling the correlation between air pollutants and these confounders is critical for causal inference. Previous studies[24] have conducted causal models by using the PM$_{2.5}$ attributable to a specific natural source (e.g., dust) as an instrumental variable that is correlated with exposure but uncorrelated with socioeconomic factors. In the present study, we showed that open fire PM$_{2.5}$ has the potential to serve as a good instrumental variable because (1) the spatiotemporal pattern of fire PM$_{2.5}$ is clearly different from that of total PM$_{2.5}$ (Fig. 1 and Supplementary Fig. 1), and (2) both fire PM$_{2.5}$ and non-fire PM$_{2.5}$ are significantly associated with health outcomes (Fig. 5). However, exploring the causal effect of PM$_{2.5}$ based on open fire exposure is beyond the scope of this study and will be explored in the next stage.

This study is limited in several ways. First, although we put some effort into controlling for potential longitudinal cofounders (e.g., adjusting the temporal trend and using bidirectional controls), the causality of the associations between open fires and pregnancy loss can be undermined if we ignore certain key temporally varying variables, such as household air pollution. Second, assessing exposure to open fire smoke is challenging, particularly in areas with poor data from ground-surface monitors. Although we applied three approaches to characterize exposure to fire smoke, we could not completely avoid the possibility of exposure misclassifications, leading to potential bias into our results. For instance, the satellite might ignore small fires, and the emissions data might be less capable of capturing long-distance transported PM$_{2.5}$ from open fires. Third, the pregnancy losses were self-reported, so the reliability of the related data should be questioned due to a few reasons (e.g., recall bias). Although the reproductive history questionnaire has been validated in preliminary studies[25,26], some outcomes could be still misclassified in the analyzed surveys. Also, uncertainty imbedded into the self-reported exposure time-window (i.e., the gestational period) impede to explore how effect of PM$_{2.5}$ varied between different trimesters. For instance, although the finding that miscarriage was more strongly associated with PM$_{2.5}$ than stillbirth suggests more adverse for the exposure during early gestational stage, the estimates were with low confidence due to potentially misclassifying the two outcomes by recalling the gestational periods (Fig. 3). Furthermore, due to the lack of relevant information in DHS, we could not distinguish self-induced abortions from pregnancy losses, which also increased the probability of misclassification of outcomes. Finally, as this study was based on selected samples that were eligible for the self-compared design, representativeness of our results should be questioned. In a previous study[27], we use a statistical simulation to show that for a homogenous true association, the self-compared design itself does not introduce bias into its estimate by selecting samples. However, our sensitivity analysis suggests the association between fire

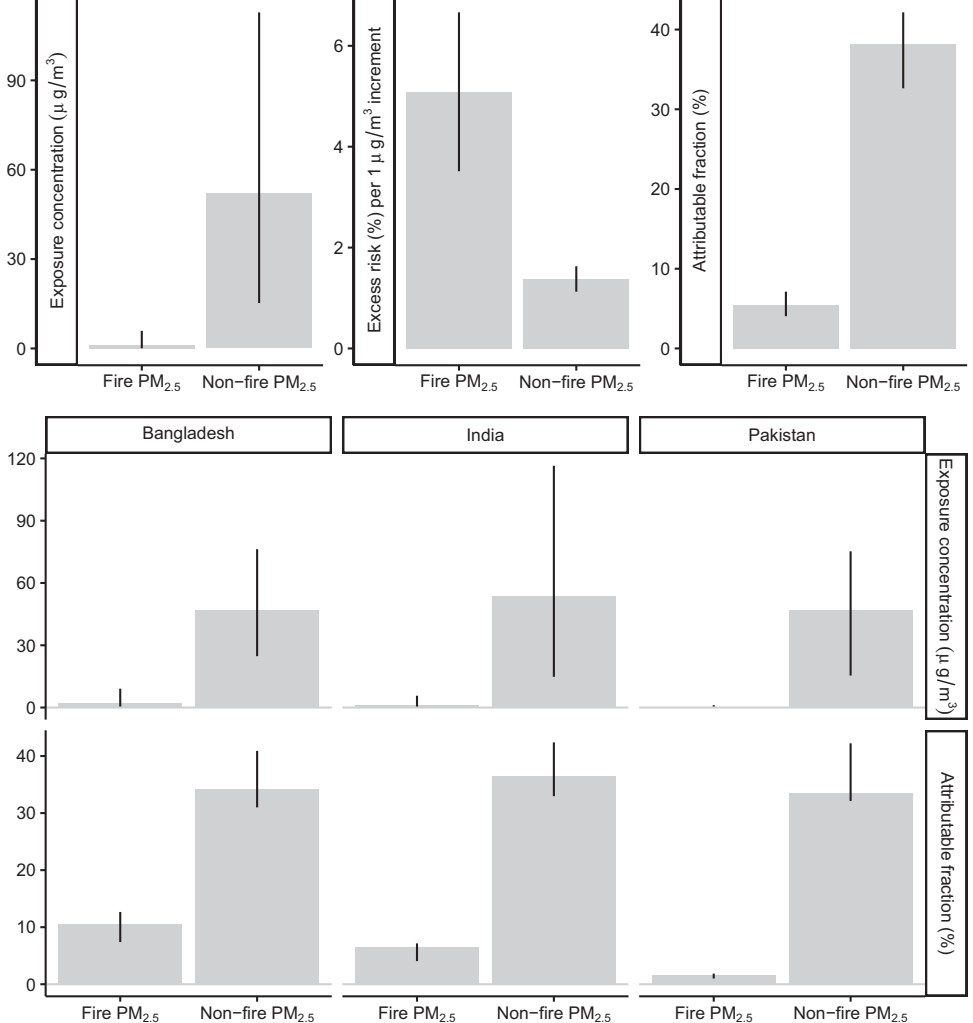

**Fig. 5 Health impacts of PM$_{2.5}$ from fire and non-fire sources.** The top panels present the comparisons between open fire and non-fire PM$_{2.5}$ in terms of exposure concentration, association magnitude, and attributable fraction. The bottom panels present the country-specific comparisons. The bars denote point-estimates, and the error bars denote the corresponding 95% confidence intervals. All estimates are derived from the full-adjusted model with 24,876 independent mothers. PM$_{2.5}$ particulate matter with a diameter of less than 2.5 μm.

PM$_{2.5}$ and pregnancy loss may be heterogenous (Fig. 3). Therefore, selecting samples (e.g., excluding samples with pregnancy losses only might lead to ignoring the younger females or the individuals who were highly susceptible to the effect of fire PM$_{2.5}$) could bias the estimate away from the average effect, and thus limited representativeness of our findings. Given the above limitations, these results should be interpreted cautiously, and it is challenging to evaluate magnitude or direction of the total bias that is jointly caused by multiple limitations. Although our previous simulation analysis[27] suggested that a combination of some limitations might result in an underestimated association, further studies with advanced designs should be performed to evaluate our findings.

In a subcontinental study in South Asia, we found a significant association between pregnancy loss and maternal exposure to fire-sourced PM$_{2.5}$. The association was not sensitive to differences in terms of covariate adjustment, inclusion criteria for controls, stratification by most of subpopulation factors (except for maternal age), but its significance level was changed by using alternative exposure indicators. Hence, we consider that the finding is moderately robust. Our study suggests that exposure to open fires can harm maternal health and contributes to a non-negligible fraction of pregnancy losses in South Asia.

## Methods

**Population data**. Individual records on women's reproductive history were collected from all available demographic and health surveys (DHS) in India, Pakistan, and Bangladesh from 2000 to 2014, when multiple exposure indicators were simultaneously available (The anthropogenic inventories[28] inputted into our CTM are until 2014, and the satellite remote sensing measurements of fire points are from 2000, as described below.). DHS is a widely used database that covers worldwide low- and middle-income countries. Each DHS is a nationally representative cross-sectional survey addressing a country's socioeconomic, demographic, and health characteristics; the survey is conducted routinely every 5 years. For more recent surveys, geographic data (i.e., the longitude and latitude for each surveyed village or residential cluster, recorded by global positioning system devices) were available and thus could be used to link the survey records with environmental variables such as air pollutants. We collected seven geocoded DHS surveys including four in Bangladesh (DHS phases 4–7), two in Pakistan (phase 5 and 7), and one in India (phase 7).

The DHS surveys are household-based instruments, and the samples were selected using a complex two-stage design. For each national survey, in the first stage, enumeration areas are selected according to census data; in the second stage, households are sampled from an updated list of households. The females of reproductive age (15–49 years) in each household were of particular interest, and their records for socioeconomic status, fertility, reproductive history, infant mortality, etc. were screened by well-trained interviewers using standard questionnaires. For eligible female participates, the response rate was varied by countries and ranged from 93 to 99% for the incorporated surveys (Supplementary Table 2). This study, and our previous study focusing on pregnancy loss in Africa, relied on the variables collected using a reproductive history questionnaire. This approach, which has been validated in preliminary studies[29], can be used to

retrospectively survey variables related to the occurrence of pregnancy termination, time and duration of the terminated pregnancy, number of normal deliveries, their birth dates, and survival status. Since some key variables such as gestational duration for a pregnancy loss were recalled by the respondents, we only selected the case that occurred mostly recent to the survey date for each mother. More details on the surveys, including the sampling approach, fieldwork procedure, and data accessibility, are documented on the official DHS website (https://www.dhsprogram.com/). The DHS data are widely employed in studies of infant and maternal health[10,30]. Procedures and questionnaires for DHS surveys have been reviewed and approved by ICF Institutional Review Board (IRB). Country-specific surveys have been approved by the ICF IRB and typically by an IRB in the host country. This study is based on the publicly available DHS data and adheres to its data usage guidelines. No further ethic approval is required.

**Exposure assessments**. Quantifying exposure levels to open fire smoke is challenging. Previous studies have used a variety of approaches, including monitoring data of ground surface total $PM_{2.5}$ on smoky days, binary satellite indicators of burned areas, binary temporal indicators of smoky or non-smoky days, and CTM-based estimates of $PM_{2.5}$ concentrations attributable to open fire[4]. There is no consensus as to which approach of exposure assessment is the best. Therefore, in this study, we used three different methods to evaluate maternal exposure to open fire smoke during gestation: satellite-based, emission-based, and CTM-based indicators of open fires, each of which has some advantages. Among the three, satellite-based indicators are the least influenced by artificial errors; emission-based measures incorporate small fires and can distinguish different fire types; and CTM-based indicators are the most easily interpreted, as they quantify exposure using $PM_{2.5}$ concentrations, which makes the effect of open fires comparable to that of common air pollutants. It is worth to highlight that those three indicators are not independent of each other. Satellite measurements are incorporated into the emissions together with other information (e.g., small fires due to agricultural waste burning), and the emissions are part of the inputs into the CTM. From satellite to emission, then to CTM, the complicated modeling procedures improve interpretability of the exposure indicator, but introduce potential artificial errors into the estimates as a cost to decouple the complexities. Since interpretability is criterial to assess the health impacts from open fires, we utilize the CTM-based indicator as main metric in our epidemiological analyses.

*Satellite remote sensing of fire points*. We first obtained the monthly global satellite product of the burned area (MCD64A1), which combines remote sensing measurements of fire by the Moderate Resolution Imaging Spectroradiometer (MODIS) from two earth-observing satellites, Terra and Aqua. Satellite remote sensors can measure the electromagnetic signals reflected from the Earth's surface, and well-developed algorithms can distinguish the signal for fires on the open surface from the radiation. An Earth-observing satellite scans the whole earth surface within a day or two and thus can capture most open fires (particularly large-scale wildfires). The product has generated valid values since 2000 at an original spatial resolution of 500 × 500 m. To spatially match the satellite indicator with other environmental variables and to quantitatively measure the likelihood of exposure to fire smoke surrounding the residences of surveyed participants, we aggregated the satellite product into a regular grid of 0.1° × 0.1° by calculating the fraction of burned area (%). Therefore, each participant was assigned to a monthly time series of burned area fraction according to his or her GPS location. The satellite product is publicly available and freely distributed by Application for Extracting and Exploring Analysis Ready Samples (https://lpdaacsvc.cr.usgs.gov/appeears/).

*Fire emissions*. The satellite-based indicator can miss some small fires such as agricultural waste burning. The recent version of global fire emission database (GFED, version 4) combines the satellite burned area data with estimated small fires[31,32] and derives a product of dry matter emission (unit: kg dry matter m$^{-2}$ month$^{-1}$) in a global grid of 0.25° × 0.25°. We obtained the monthly GFED data during 2000–2014 from the GFED website (http://globalfiredata.org/) and directly used the emission data in the pixel corresponding to a participant's geographic location as an indicator of open fire exposure. Furthermore, GFED partitions the dry matter emissions by contributions from different types of fire: (1) savanna, grassland, and shrubland fires; (2) boreal forest fires; (3) temperate forest fires; (4) deforestation and degradation; (5) peatland fires; and (6) agricultural waste burning. Among these, fires of types 1, 3, 4, and 6 can be found in our study domain. We also considered the type-specific dry-matter emissions as an exposure indicator.

*$PM_{2.5}$ concentrations attributable to open fires*. Using emission data directly can underestimate fire smoke exposure contributed by burning plumes transported over long distances. To address this problem, we estimated the $PM_{2.5}$ concentrations attributable to open fire emissions using a global chemical transport model (CTM), GEOS-Chem. GEOS-Chem is a widely used CTM that has been applied to estimate the contribution to ambient air pollutant concentrations from a specific emission sector. A similar approach has been used in epidemiological studies to assess fire exposure[33].

In this study, we used the GEOS-Chem version 11-01 driven by assimilated meteorological fields from the NASA Global Modeling and Assimilation Office's Modern-Era Retrospective analysis for Research and Applications Version 2 (MERRA-2). The gridded product has a spatial resolution of 0.5° × 0.625° and can be freely accessed from the website of the NASA Goddard Earth Sciences Data and Information Services Center (https://disc.gsfc.nasa.gov/). The GEOS-Chem model has a spatial resolution of 2° × 2.5° and 47 vertical layers. The model was run with the full $O_x - NO_x - CO - VOC - HO_x$ chemistry, which includes sulfate-nitrate-ammonium, primary and secondary carbonaceous aerosols, mineral dusts and sea-salts. Specifically, sulfate-nitrate-ammonium is simulated through the ISORROPIA-II thermo-dynamical equilibrium[34]. The aerosol simulations have been extensively evaluated using measurement data[35–37]. The global anthropogenic emission inventory Community Emissions Data System (CEDS)[28] was used to drive the GEOS-Chem model, and the fire emissions were taken from GFED4. We conducted GEOS-Chem simulations from 2000–2014 with a six-month spin-up starting from July 1999. $PM_{2.5}$ in the bottom layer were taken to represent the ambient $PM_{2.5}$ concentrations.

The GEOS-Chem simulated $PM_{2.5}$ concentrations over our study domain were evaluated by (1) ground-surface monitoring data from five Indian cities (i.e., Chennai, Delhi, Hyderabad, Kolkata, and Mumbai), and (2) the satellite-based $PM_{2.5}$ estimates. The monitoring data were obtained from the U.S. Embassy and Consulates in India (https://in.usembassy.gov/embassy-consulates/new-delhi/air-quality-data/), and were aggregated as the monthly averages during 2013–2014. The GEOS-Chem simulations were in good agreement with the in-situ observations (Pearson's $R^2 = 0.69$, Supplementary Fig. 4), the gold-standard values for exposure assessment. Since the spatiotemporal coverage of monitoring data was limited, we further compared the GEOS-Chem simulations with the satellite-based $PM_{2.5}$ estimates[38], which were obtained from a global annual product during 2000–2014 with a spatial resolution of 0.1° × 0.1°. The satellite-based $PM_{2.5}$ estimates[38], which have been reported to be in good correlation (Pearson's $R^2 = 0.81$) with ground-surface concentrations, were moderately correlated with annual averages of our simulations (Pearson's $R^2$ varied each year from 0.47 to 0.60, Supplementary Table 3).

To calculate the fire-induced $PM_{2.5}$ fractions, we conducted two GEOS-Chem simulation scenarios, one with and one without fire emissions. The fractional contribution of open fire to simulated $PM_{2.5}$ ($\rho_m$) can be calculated as:

$$\rho_{m,y} = (PM_{2.5,m,y}^{\text{wfire}} - PM_{2.5,m,y}^{\text{nofire}}) / PM_{2.5,m,y}^{\text{wfire}} \quad (1)$$

where the superscripts wfire and nofire represent model simulations with and without fire emissions, respectively; and subscripts $m$ and $y$ represent the month and year indices, respectively. Additionally, considering the uncertainties in emission inventories and modeling procedures (such as parametrization of chemical reactions, and model boundary conditions)[39], bias correction with ground-surface observations has been widely applied to improve the performance of CTM in exposure assessment[40]. Therefore, we introduced the annual mean of satellite-based $PM_{2.5}$ estimates[38] ($PM_{2.5,y}^{\text{satellite}}$), which have used in our previous studies[10,27] exploring the association between total $PM_{2.5}$ and pregnancy loss, as a constraint to calibrate our GEOS-Chem outputs. The calibration rate can be calculated as:

$$\eta_y = PM_{2.5,y}^{\text{satellite}} / (1/12 \times \sum_m PM_{2.5,m,y}^{\text{wfire}}) \quad (2)$$

In this way, monthly exposure metrics were calculated as:

$$[\text{Total } PM_{2.5}]_{m,y} = \eta_y \times PM_{2.5,m,y}^{\text{wfire}};$$
$$[\text{Fire } PM_{2.5}]_{m,y} = \eta_y \times \rho_{m,y} \times PM_{2.5,m,y}^{\text{wfire}}; \quad (3)$$
$$[\text{Non-fire } PM_{2.5}]_{m,y} = \eta_y \times (1 - \rho_{m,y}) \times PM_{2.5,m,y}^{\text{wfire}}$$

In the above equations, [Total $PM_{2.5}$] denotes the total concentration of all $PM_{2.5}$ components, [Fire $PM_{2.5}$] denotes the concentration of $PM_{2.5}$ particles attributable to emissions from open fire, and [Non-fire $PM_{2.5}$] denotes the concentration of $PM_{2.5}$ particles attributable to the other sources. We downscaled the GEOS-Chem outputs ($PM_{2.5,m,y}^{\text{wfire}}$, $PM_{2.5,m,y}^{\text{nofire}}$ and $\rho_{m,y}$) to the 0.1° × 0.1° grid using an inverse-distance-weighting approach, to match them with the satellite-based $PM_{2.5}$ estimates. After the calibration, the performance of the GEOS-Chem simulations ($PM_{2.5,m,y}^{\text{wfire}}$) was improved, according to all statistical metrics of the comparison with monitoring data (Supplementary Fig. 4). For instance, the mean bias was reduced from 26.6 to 20.8 μg/m$^3$ after the calibration.

After data preparation, we assigned monthly series of open fire $PM_{2.5}$ concentrations and non-fire $PM_{2.5}$ concentrations to each participant, through spatially matching the residential address (longitude and latitude) geocoded by DHS with the corresponding pixel in the gridded $PM_{2.5}$ maps. For comparative purpose, the alternative exposure indicators (i.e., fire emissions and satellite images) were prepared in the same way as the $PM_{2.5}$ preparation.

**Epidemiological design and statistical analyses**. To evaluate the association of maternal exposure to open fire during gestation with the probability of pregnancy loss, we applied a previously established self-comparison case–control design[10]. From the original database, we extracted pairs including a pregnancy loss case and matched control(s) in the form of a successful delivery by the same mother. This matching approach can be used to control for risk factors that vary inter-individually but rarely in a temporal dimension, such as genetics, and is thus a cost-

 

efficient way to adjust for complex confounders in a large population study[41] in which the individuals can be highly heterogeneous. In such a self-comparison study, a mother who reports greater exposure associated with a prematurely terminated delivery compared with a normal delivery serves as evidence to support a positive association between exposure and increased risk of pregnancy loss. In the comparison, we further matched the exposure time window between cases and their corresponding controls. We calculated gestational exposure by averaging environmental variables (i.e., fire $PM_{2.5}$, non-fire $PM_{2.5}$, temperature and humidity) from the month of conception to the month of pregnancy termination, which were recalled by the mothers; for the matched control, we calculated average exposure during the same gestational period (rather than the whole duration of gestation for the control birth).

According to this epidemiological design, we only selected mothers who reported a case of pregnancy loss and at least one successful delivery. Of the 782,918 female respondents to the seven surveys, 102,427 were cases of pregnancy loss. Of these, 31,303 losses occurred within our study period, i.e., 2000–2014. After excluding 6427 cases who could not be matched with eligible controls (i.e., successful deliveries with valid exposure assessments), the analysis finally involved 24,876 mothers. In total, they reported 75,262 delivery events, including the most recent case and all available controls for each mother.

According to the matched design, we used a conditional logit regression to quantify the association between indicators of open fire and pregnancy loss. The adjusted covariates included maternal age; spline terms for temperature (with 3 degrees of freedom [DF]), humidity (DF = 3), and the month of conception (DF = 4); and a categorical term for year of conception. The temperature and humidity splines were employed to control for nonlinear effects of climate, and the month spline to model the periodic variation in pregnancy loss. Since maternal age has been associated with the increased risk of pregnancy loss in a nonlinear relationship[42], it was modeled as a categorical variable (i.e., maternal age <20, 20–24, 25–29, 30–34, 35–39, or ≥40 years). We used a set of calendar-year-specific intercepts to further control for the long-term trend (e.g., improvement of maternal health driven by economic development) in the outcome. The regression models were further mutually adjusted by non-fire $PM_{2.5}$ to explore whether the open fire effect was confounded by $PM_{2.5}$ from other sources. The association was evaluated using the OR and 95% confidence interval (CI) for each unit increment in exposure (i.e., 1 μg/m³ $PM_{2.5}$, 1% satellite burned area, or 10 g/m³/month dry-matter emission). To evaluate the impact of open fire or non-fire $PM_{2.5}$, we also calculated the attributable fraction (AF) according to the equation: AF = 1 − 1 / exp [β × ($PM_{2.5}$ − TMREL)], where $PM_{2.5}$, TMREL, and β denote the mean exposure level in the analyzed samples, the theoretical minimum risk exposure level, and the corresponding regression coefficient, respectively. We set the TMREL as 10 μg/m³ for non-fire $PM_{2.5}$ exposure according to the air quality guidelines of World Health Organization and as 0 μg/m³ for the open fire $PM_{2.5}$ exposure.

**Sensitivity analyses.** We ran multiple sensitivity analyses to examine the robustness of the association between open fire $PM_{2.5}$ and pregnancy loss. First, we tested the homogeneity of the association among subpopulations by exploring the interaction between open fire $PM_{2.5}$ and subgroup indicators (e.g., education level). Second, we re-evaluated the association in a subset of the samples by refining the inclusion criteria for the cases or controls. We divided the cases into miscarriages or stillbirths according to their gestational duration (miscarriage: gestation <5 months; stillbirth: gestation ≥5 months) and then constructed two corresponding subsets. Two other subsets were selected by restricting the controls to singleton births or to healthy newborns who survived more than 12 months. In the subset defined by parity, either cases or controls were required to be nulliparous or multiparous (i.e., we excluded the case–control pairs which were not matched up in terms of parity). For every case–control pair in the nulliparous subset, the first successful delivery (i.e., control) occurred following the corresponding pregnancy loss (i.e., case). Third, due to the lack of control over longitudinal confounders in this self-comparison case–control design, the estimated association could be biased by long-term trends in the baseline risk of pregnancy loss. Matching a case with controls both before and after the time of pregnancy loss can help to reduce this type of bias. We termed this approach "bidirectional control," referring to a similar method used in case-crossover studies[43]. Fourth, open fires might harm maternal health through pathways other than ambient exposure, such as directly destroying their dwelling. To examine whether airborne smoke was the major variable to explain the association between open fire and pregnancy loss, we defined exposure to transported fire $PM_{2.5}$ by restricting the samples (cases or controls) located in the places, where the satellite indicator of the burned area was equal to zero. Fifth, although the retrospective reproductive questionnaire has been evaluated and validated by experimental studies before surveys[25,26], our estimates might still be limited by the potential recall bias in DHS variables. To explore that, we selected the cases and controls with recall periods less than $n$ years ($n$ = 3, 4, 5, or 6), and re-estimated the associations. Sixth, we explored the nonlinear association between pregnancy loss and open fire $PM_{2.5}$ by replacing the linear term with a smoothing spline term. Finally, we derived the association between fire emission and pregnancy loss for specific types of fire (i.e., savanna, grassland, and shrubland fires; temperate forest fires; deforestation and degradation; and agricultural waste burning).

**Reporting summary**. Further information on research design is available in the Nature Research Reporting Summary linked to this article.

## Data availability

The Demographic and Health Survey (DHS) data, satellite fire data (MODIS MCD64A1), global fire emission database (GFED, version 4), anthropogenic emission inventory of Community Emissions Data System (CEDS), Modern-Era Retrospective analysis for Research and Applications Version 2 (MERRA-2) data, and satellite-based $PM_{2.5}$ data that support the findings of this study are available from https://www.dhsprogram.com/, https://lpdaacsvc.cr.usgs.gov/appears/, http://globalfiredata.org/, https://esgf-node.llnl.gov/search/input4mips/, https://disc.gsfc.nasa.gov/, and http://fizz.phys.dal.ca/~atmos/martin/?page_id=140, respectively. The fire $PM_{2.5}$ data that supporting the findings of this study are derived from the GFED, CEDS, MERRA-2, and satellite-based $PM_{2.5}$ data using a standard model of GEOS-Chem (version 11-01, freely available from http://acmg.seas.harvard.edu/geos/), and are available from the corresponding author upon reasonable request. The ground-surface monitoring data to evaluate GEOS-Chem simulations are from https://in.usembassy.gov/embassy-consulates/new-delhi/air-quality-data/. Source data for the figures are provided within supplementary data. Source data are provided with this paper.

## Code availability

The raw R codes for the epidemiological analyses are documented in the supplemental files. The R codes and relevant data to reproduce the figures are also within supplemental files.

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

## Acknowledgements

This work was supported by National Natural Science Foundation of China (41701591 and 81571130100), Ministry of Science and Technology of China (2015CB553401), and CAMS Innovation Fund for Medical Sciences (Grant No. 2017-I2M-1-004).

## Author contributions

T.X. & T.Z. designed the study; T.X., H.W., and J.L. prepared the health data; G.G. performed the GEOS-Chem simulations and prepared the exposure data; T.X., G.G., H.W., and J.L. analyzed the data; T.X., Y.H., H.W., J.L., H.L., Y.Z., and T.Z interpreted the results together; T.X., G.G., and Y.H. prepared the first draft; T.X. and T.Z. supervised the study; all co-authors revised the manuscript together.

## Competing interests

The authors declare no competing interests.
