## [Peer Review File · Nature Communications]

Reviewers' Comments:

Reviewer #1:

Remarks to the Author:

This paper investigates wildfire smoke-associated pregnancy losses in south Asia. I applaud the authors for investigating this under-studied pregnancy endpoint, and for using a strong study design with a comprehensive suite of sensitivity analyses.

The majority of my comments are minor and probably reflect lack of detail rather than incorrect methods or interpretation. However, I do think the authors oversell the consistency and robustness of their results to some extent, and I encourage them to be more circumspect. That being said, this is a strong paper on a very important topic.

Line 56 – I assume the point estimate units are $1\mu\text{g}/\text{m}^3$ daily averaged over the entire gestational period? Please describe the time frame the exposure covers. A $1\mu\text{g}/\text{m}^3$ daily increase over a month or a trimester is a very different cumulative exposure than a $1\mu\text{g}/\text{m}^3$ daily increase over a pregnancy.

Also in reference to the point estimate, I assume this is from the modeled PM_{2.5} method, not the others? The authors introduce three exposure characterization methods but then only focus on one, the CTM. Either present the CTM analysis as the primary analysis and shift the others to sensitivity analyses or present all three equally.

Line 105 – Is this sentence: “together contributing 105 approximately one-third (35.3%) of all stillbirths despite having only one-quarter (25.8%) of all 106 newborns.” comparing to all still births everywhere in the world? Or is it saying that 35.3% of stillbirths are spontaneous abortions? I’m confused. If the former, please say “of all stillbirths in the world”.

Line 197 – typo in the equation.

Line 201 – Why is the calibration necessary? What sort of biases does the CTM potentially have? Why is an annual calibration better than a monthly calibration? Is the satellite data any better? Which satellite-based estimate are you referring to, anyway? Are you referring to the point emissions database GFED? I wouldn't thinking an emissions database would accurately ground-truth for ambient concentrations.

Line 226 – Does the health database include information on date of conception? Or last period? How likely are women to remember the date of last period for a pregnancy that may have taken place years in the past? What is the risk of recall bias or errors? Are the women's memories backed-up by medical records? How many years in the past does the health survey ask about? How were the control/referent births selected if there were more than one?

Line 237 – Did you include all possible control births or just choose one from each woman?

Line 240 -- How was maternal age modeled? Categorical, linear, or nonlinear? Linear probably wouldn't make sense.

Line 240 – Were temp and humidity averaged over the gestation period? Please be clear.

Line 268 – There are known biases with bidirectional designs.

Figure 1b is a bit confusing. The color scheme makes it appear that the white band in the middle is what is important. I suggest using a black/white background (+theme_bw()) and two different greys for the wildfire and non-wildfire PM.

Line 319 – I disagree with the statement “These results were not sensitive to covariate adjustments or mutual adjustments with non-fire PM_{2.5}.” First, this is a judgement call that

belongs in the discussion, not the results. Second, It's not even true; two of the fire characterization methods switch between consistent with the null and inconsistent with the null depending on which covariates are in the model.

Line 333/334 – This is an interesting finding. Please present the numerical results.

Line 337 --- I don't know what "considerably changed" means. Be more precise.

Line 348 – Could you perform a statistical test comparing the nonlinear exposure/response model to the linear model?

Line 356 – Is this using the CTM method? Are these estimates adjusted for covariates and non-smoke PM?

Figure 3 – Please put the legend outside the plot. Same comment for Figure S3.

Figure 4 – This is an enormous OR at the high end of exposure. More discussion of this result is warranted.

Line 370 – Nothing in this paper is testing toxicity; I would avoid using this term.

Line 371 – The estimates state here do not match those in the rest of the paper (1.4% compared to 5.1% in Table 1).

Line 374/375 – Where would I find the numbers to support this claim? Figure 5?

Lines 411-413 – "Among various adverse birth outcomes, pregnancy loss may be the most dangerous because it not only reflects the most severe damage to the fetus but also will increase the risk of other outcomes in subsequent deliveries" If this is the case you should probably adjust the statistical models for whether the pregnancy loss occurred before or after the control.

Line 417 – Again, you haven't studied toxicity in the biological sense. Avoid using the term "toxicity" in reference to this work.

Line 449-451 – The statement "The estimated association was not sensitive to differences in terms of covariate adjustment, subpopulation stratification, inclusion criteria for controls, and exposure assessment indicators" is simply not true. The CTM results are clearly sensitive to maternal age, and the results using other exposure metrics yield different answers depending on which covariates are used. Please be more careful in your wording.

Reviewer #2:

Remarks to the Author:

In "Open fire exposure increases the risk of pregnancy loss in South Asia," the authors effectively describe findings from a study investigating associations between fires and pregnancy loss. This is an outstanding manuscript of high public health importance. The methods are clearly described, comprehensive, and innovative. The sensitivity analyses are well-justified and described. The results point to the critical importance of reducing exposure to air pollution during pregnancy. The following minor revisions are suggested.

Introduction

1. Suggest that the authors use "participants" instead of "subjects" throughout the manuscript.
2. Minor typo is "risk" in first paragraph.

Methods

1. The authors provide a link and relevant references to the DHS surveys in the first two paragraphs of the Methods. However, it would be helpful to briefly mention response rates and the sampling approach. For example, is the DHS able to reach all populations within the countries of interest. Are there particular populations that were excluded or difficult to reach?
2. The spatial area or buffer is not clear for the exposure metrics derived from satellite remote sensing and fire emissions data. For example, within what area does the percent burned area apply?
3. The authors examine effect modification by parity in sensitivity analyses. Why not also examine parity as a potential confounder? Did the authors consider time between births as well?

Results

1. Figure 1. Please define abbreviations.
2. First paragraph: Please describe how you defined "rural."
3. Save last sentence of "Associations by different exposure indicators" section for Discussion.
4. Save last sentence of second to last paragraph of "Sensitivity analyses..." section for Discussion.

Discussion

1. Please provide additional discussion of the following:
 - a. What information might be lost by excluding women who experienced pregnancy loss but no live births?
 - b. Differences in toxicity of different types of open fires.
 - c. The importance of timing of exposure during pregnancy. For example, were women more sensitive during the third trimester?
2. In the last paragraph of the Discussion, please comment on the likely direction of the bias that could result from the limitations described.

Reviewer #3:

Remarks to the Author:

General Description:

The authors use health data and 3 indicators of PM_{2.5} associated with fires to determine health risk to pregnant women of exposure to PM_{2.5} from fires in South Asia. The results from this work and the focus region will be of general interest to a wide audience. I would recommend publication, after providing additional details about the methods used and results obtained. These are specified below.

General Comments:

Climate change is invoked throughout the manuscript as the cause for trends and variability in open fires in South Asia, but it is well known that severely degraded air quality in northern India is due to burning of agricultural waste (see for example: <https://doi.org/10.1038/s41598-019-52799-x>).

Is it surprising that the 3 metrics used give corroborative results, given that these datasets are not independent? GEOS-Chem uses the GFED inventory for open fire emissions, and GFED uses satellite-derived burned area to determine the location and extent of fires.

The description of the model includes superfluous information ("can be freely accessed from the website..." when referring to the MERRA-2 meteorology) and is missing relevant details. The latter includes the domain sampled (should be global for the coarse resolution given), the time range that the model was simulated, the amount of time the model was spun up before sampling, and how the relevant primary and secondary aerosol components are represented in the model.

The papers that the authors reference in support of model evaluation are from much earlier

versions of the model and don't necessarily evaluate the model over the study region.

In the description of the model calibration: What satellite-based product do the authors use? Include an appropriate reference for this. At what spatial resolution is the calibration conducted? Does the calibration typically correct for a low or high bias in the model? What is the size of this bias?

Why does a small fire PM2.5 contribution of 2% result in a large impact on health (13% of disease burden associated with PM2.5)?

Specific Comments:

Abstract: consider replacing "pre-established database" with "inventory" and "referent" with "reference".

Methods description of Population data: In the first paragraph "CTM-based open fires PM2.5 concentrations were available" is provided as the reason for focusing on 2000 to 2014, but the model isn't limited to these years.

In the paragraph starting "Table 1 presents the associations...", what does "model adjustments" mean? In the last sentence of this paragraph, could the very coarse spatial resolution of the model be a contributing factor?

In Fig. 2, what does "P = " represent?

In the Discussion, what is "common PM2.5"?

To reviewers,

We thank the reviewers for the detailed and professional comments. The responses are listed point-by-point as follows. The revisions corresponding to specific comments are highlighted by blue color in our responses below.

Reviewer #1

[General comments] This paper investigates wildfire smoke-associated pregnancy losses in south Asia. I applaud the authors for investigating this under-studied pregnancy endpoint, and for using a strong study design with a comprehensive suite of sensitivity analyses.

The majority of my comments are minor and probably reflect lack of detail rather than incorrect methods or interpretation. However, I do think the authors oversell the consistency and robustness of their results to some extent, and I encourage them to be more circumspect. That being said, this is a strong paper on a very important topic.

[Response] We thank the reviewer for the positive comments on our manuscript. We revise the conclusions and improve their precision according to the reviewer's detailed comments below.

[Specific comments]

1. Line 56 – I assume the point estimate units are 1 ug/m^3 daily averaged over the entire gestational period? Please describe the time frame the exposure covers. A 1 ug/m^3 daily increase over a month or a trimester is a very different cumulative exposure than a 1 ug/m^3 daily increase over a pregnancy.

[Response] Thank for the reminder. According to the comment, we revise the sentence as “The adjusted odds ratio (OR) of pregnancy loss for a $1\text{-}\mu\text{g/m}^3$ increment in averaged concentration of fire $\text{PM}_{2.5}$ during pregnancy was estimated as 1.051 (95% confidence intervals [CI]: 1.035, 1.067).” (L22-24). The exposure was quantified during the entire period between conception and termination of pregnancy, and thus $1\text{-}\mu\text{g/m}^3$ increment for the fire-sourced $\text{PM}_{2.5}$ is a considerable amount of exposure.

2. Also in reference to the point estimate, I assume this is from the modeled $\text{PM}_{2.5}$ method, not the others? The authors introduce three exposure characterization methods but then only focus on one, the CTM. Either present the CTM analysis as the primary analysis and shift the others to sensitivity analyses or present all three equally.

[Response] Suggestion accepted. We revise the relevant illustration of the method as “Exposure was assessed using a chemical transport model as the concentrations of fire-sourced $\text{PM}_{2.5}$ (i.e., fire $\text{PM}_{2.5}$).” (L21-22). Here, we mention the CTM as the primary exposure indicator. Limited by the space of abstract (< 150 words), we don't have space to describe the alternative methods as sensitivity analyses in abstract.

3. Line 105 – Is this sentence: “together contributing approximately one-third (35.3%) of all stillbirths despite having only one-quarter (25.8%) of all newborns.” comparing to all still births everywhere in the world? Or is it saying that 35.3% of stillbirths are spontaneous abortions? I'm confused. If the former, please say “of all stillbirths in the world”.

[Response] Suggestion accepted. Accordingly, we modify the sentence as “we conducted an epidemiological study in three South Asian countries, India, Pakistan, and Bangladesh, which have the highest rates of pregnancy loss in the world¹⁶, together contributing approximately one-third (35.3%) of all stillbirths in the world despite having only one-quarter (25.8%) of all newborns.” (L64-68).

4. Line 197 – typo in the equation.

[Response] Thank for pointing out the typo. We correct it in the revised version.

5. Line 201 – Why is the calibration necessary? What sort of biases does the CTM potentially have? Why is an annual calibration better than a monthly calibration? Is the satellite data any better? Which satellite-based estimate are you referring to, anyway? Are you referring to the point emissions database GFED? I wouldn't thinking an emissions database would accurately ground-truth for ambient concentrations.

[Response] First, we didn't refer to GFED data when mentioning “satellite-based $\text{PM}_{2.5}$ ”.

Second, to evaluate the bias in CTM, we add a few analyses, which are described as “The GEOS-Chem simulated PM_{2.5} concentrations over our study domain were evaluated by (1) ground-surface monitoring data from five Indian cities (i.e., Chennai, Delhi, Hyderabad, Kolkata, and Mumbai), and (2) the satellite-based PM_{2.5} estimates. The monitoring data were obtained from the U.S. Embassy and Consulates in India (<https://in.usembassy.gov/embassy-consulates/new-delhi/air-quality-data/>), and were aggregated as the monthly averages during 2013-2014. The GEOS-Chem simulations were in good agreement with the in-situ observations ($R^2 = 0.69$, Figure S4), the gold-standard values for exposure assessment. Since the spatiotemporal coverage of monitoring data was limited, we further compared the GEOS-Chem simulations with the satellite-based PM_{2.5} estimates²⁷, which were obtained from a global annual product during 2000–2014 with a spatial resolution of $0.1^\circ \times 0.1^\circ$. The satellite-based PM_{2.5} estimates²⁷, which have been reported to be in good correlation ($R^2 = 0.81$) with ground-surface concentrations, were moderately correlated with annual averages of our simulations (R^2 was varied by year from 0.47 to 0.60, Table S3).” (L347-359).

Third, to illustrate the purposes for the calibration and to clarify data sources, we add a few sentences. Now the revised words read as “Additionally, considering the uncertainties in emission inventories and modeling procedures (such as parametrization of chemical reactions, and model boundary conditions)²⁸, bias correction with ground-surface observations has been widely applied to improve the performance of CTM in exposure assessment²⁹. Therefore, we introduced the annual mean of satellite-based PM_{2.5} estimates²⁷ (PM_{2.5, y}^{satellite}), which have used in our previous studies^{10, 30} exploring the association between total PM_{2.5} and pregnancy loss, as a constraint to calibrate our GEOS-Chem outputs.” (L366-371) and “After the calibration, the performance of the GEOS-Chem simulations (PM_{2.5, m, y}^{wfire}) was improved, according to all statistical metrics of the comparison with monitoring data (Figure S4). For instance, the mean bias was reduced from 26.6 to 20.8 $\mu\text{g}/\text{m}^3$ after the calibration.” (L383-385).

Finally, we didn’t perform the monthly calibration for two reasons: (1) the satellite-based concentrations of PM_{2.5} were only evaluated at annual scale, we didn’t know how it performed on monthly scale; and (2) the open-accessed version of the product was in annual scale. We didn’t mean the annual calibration was better than monthly calibration, but it seems to be the best-available option for our study domain and period.

6. Line 226 – Does the health database include information on date of conception? Or last period? How likely are women to remember the date of last period for a pregnancy that may have taken place years in the past? What is the risk of recall bias or errors? Are the women’s memories backed-up by medical records? How many years in the past does the health survey ask about? How were the control/referent births selected if there were more than one?

[Response] Yes, DHS contained enough data to characterize the period of effective exposure (from conception to pregnancy termination). To enhance the relevant illustration, we add a few words. Now the relevant sentences now read as “This approach, which has been validated in preliminary studies¹⁸, can be used to retrospectively survey variables related to the occurrence of pregnancy termination, time and duration of the terminated pregnancy, number of normal deliveries, their birth dates, and survival status.” (L272-275) and “we further matched the exposure time window between cases and their corresponding controls. We calculated gestational exposure by averaging environmental variables (i.e., fire PM_{2.5}, non-fire PM_{2.5}, temperature and humidity) from the month of conception to the month of pregnancy termination, which were recalled by the mothers; for the matched control, we calculated average exposure during the same gestational period (rather than the whole duration of gestation for the control birth).” (L401-406).

For the recall bias, we agree with the reviewer that it is a potential limitation in our study. The memories were not backed up by medical records and thus the recall bias could not be completely avoided. Accordingly, we do the following two modifications.

- We point out the recall bias as the potential limitation in our study. The revised

sentence now reads as “This study is limited in several ways. Third, the pregnancy losses were self-reported, so the reliability of the related data should be questioned due to a few reasons (e.g., recall bias).” (L218-220).

- We conduct a new sensitivity analysis to examine how the recall bias affects the estimated associations. To illustrate that, in method section, we add a few words, which read as “Fifth, although the retrospective reproductive questionnaire has been evaluated and validated by experimental studies before surveys^{34, 35}, our estimates might still be limited by the potential recall bias in DHS variables. To explore that, we selected the cases and controls with recall periods less than n years ($n = 3, 4, 5, \text{ or } 6$), and re-estimated the associations.” (L454-457). To describe the results, we add a sentence, which reads as “restricting the analyzed samples by different recall periods didn’t change the significance level of the estimated associations” (L191-120). The updated results are presented in new Figure 3.

The figure also contains information on how many events occurred recently to the survey date. For instance, 15,024 out of 24,876 pregnancy losses (~ 60%) were in the past five years before survey.

For a mother with multiple cases, we only selected the most recent one. We selected all available controls in the main analyses. To illustrate that, we add a few sentences, which read as “Since some key variables such as gestational duration for a pregnancy loss were recalled by the respondents, we only selected the case that occurred mostly recent to the survey date for each mother.” (L275-277), and “the analysis finally involved 24,876 mothers. In total, they reported 75,262 delivery events, including the most recent case and all available controls for each mother.” (L411-414).

7. Line 237 — Did you include all possible control births or just choose one from each woman?
[Response] Yes, we involved all possible controls. For detailed revisions, please refer to the response to above comment. For some subset sensitivity analyses, such as “Case–control: nulliparous” (Figure 3), we used a one-vs-one matched case-control design.
8. Line 240 -- How was maternal age modeled? Categorical, linear, or nonlinear? Linear probably wouldn’t make sense.
[Response] We did model the age as a categorical variable, as reviewer suggested. To illustrate that, we add a new sentence, which reads as “Since maternal age has been associated with the increased risk of pregnancy loss in a nonlinear relationship³², it was modelled as a categorical variable (*i.e.*, maternal age < 20, 20 ~ 24, 25 ~ 29, 30 ~ 34, 35 ~ 39, or ≥ 40 years).” (L419-421).
9. Line 240 – Were temp and humidity averaged over the gestation period? Please be clear.
[Response] Yes. We add a few words to clarity that. The revised sentences now read as “We calculated gestational exposure by averaging environmental variables (*i.e.*, fire PM_{2.5}, non-fire PM_{2.5}, temperature and humidity) from the month of conception to the month of pregnancy termination, which were recalled by the mothers; for the matched control, we calculated average exposure during the same gestational period (rather than the whole duration of gestation for the control birth).” (L420-406).
10. Line 268 – There are known biases with bidirectional designs.
[Response] Suggestion accepted. Here, we aimed to mean that bidirectional design was more capable to control for the bias from long-term temporal trend than the unidirectional design. We agree with the reviewer that bidirectional design cannot completely avoid bias. Accordingly, we revised the relevant sentence to make it more appropriate. Now the relevant sentence reads as “Matching a case with controls both before and after the time of pregnancy loss can help to reduce this type of bias.” (L447-448).

11. Figure 1b is a bit confusing. The color scheme makes it appear that the white band in the middle is what is important. I suggest using a black/white background (+theme_bw()) and two different greys for the wildfire and non-wildfire PM.
[Response] We thank the reviewer for the very detailed and specific suggestion. We modify the Figure 1b, as suggested.
12. Line 319 – I disagree with the statement “These results were not sensitive to covariate adjustments or mutual adjustments with non-fire PM_{2.5}.” First, this is a judgement call that belongs in the discussion, not the results. Second, It’s not even true; two of the fire characterization methods switch between consistent with the null and inconsistent with the null depending on which covariates are in the model.
[Response] Suggestion accepted. Here, the original sentence only means the models on Fire PM_{2.5} were robust, but it did cause some ambiguities. As we mentioned in results, “The direction of the estimated association between fire and pregnancy loss was not changed by using different indicators of exposure.” (L98-100) and “However, the significance level of the ORs estimated by satellite data on burned area or dry-matter emissions data was sensitive to model adjustments.” (L101-102). Accordingly, we remove the sentence in results section.
13. Line 333/334 – This is an interesting finding. Please present the numerical results.
[Response] Suggestion accepted. To illustrate the results, we add a new sentence, which reads as “For mothers with a maternal age < 30, 30 ~ 34, or ≥ 35 years, each increment of 1 µg/m³ PM_{2.5} was associated with an excess pregnancy-loss risk of 4.1% (95% CI: 2.4%, 5.8%), 7.6% (95% CI: 4.0%, 11.3%) or 11.1% (95% CI: 5.5%, 17.0%), respectively.” (L110-113).
14. Line 337 --- I don’t know what “considerably changed” means. Be more precise.
[Response] Suggestion accepted. We modify the sentence as “Figure 3 shows that the direction of the association between fire PM_{2.5} and pregnancy loss was not changed when different inclusion criteria were applied to the samples.” (L115-116).
15. Line 348 – Could you perform a statistical test comparing the nonlinear exposure/response model to the linear model?
[Response] Suggestion accepted. We applied a likelihood ratio test to examine the nonlinearity. The p-value is reported as < 2.2e-16. To state the relevant results, we add a few words, which read as “To examine the hypothesis of linear association, we used a likelihood ratio test, which compared the nonlinear model with the fully-adjusted linear model. The test result indicated a nonlinear association (P-value < 0.001).” (L126-128).
16. Line 356 – Is this using the CTM method? Are these estimates adjusted for covariates and non-smoke PM?
[Response] Yes. To illustrate that, in the relevant figure captions (Figures 2-4), we add a few words, which read as “Concentrations of fire PM_{2.5} were assessed by the GEOS-Chem model, and the associations were adjusted by maternal age, temperature, humidity, temporal trends and non-fire PM_{2.5}.” (L602-603).
17. Figure 3 – Please put the legend outside the plot. Same comment for Figure S3.
[Response] Suggestion accepted. The relevant figures (Figures 3 and S3) are modified, accordingly.
18. Figure 4 – This is an enormous OR at the high end of exposure. More discussion of this result is warranted.
[Response] Suggestion accepted. We add a few sentences to enhance our discussion on the relevant findings. In the new materials, we particularly highlight the limitations of the findings. Now the revised sentences read as “We also found a super-linear exposure–

response between fire PM_{2.5} and pregnancy loss (Figure 4). The result suggests that the health impacts from large open fires, such as forest fires, should be considered. Our analysis of type-specific associations between fire emissions and pregnancy loss showed a consistent result (Figure S3). Stronger associations were found for the large fires (*i.e.*, grassland, shrubland fires, and temperate forest fires) compared to the small fires (*i.e.*, deforestation and degradation). However, the interpretation should be cautious for two reasons: first, by quantifying the exposure using fire PM_{2.5} concentrations, our analyses ignored the complex behavior of individual fires, which can be different in terms of size, duration and speed⁴⁰. All dimensions of fire behavior can affect the exposure pattern and chemical species of fire smoke, and thus the relevant health effect. Second, since frequency for large fires is low, our estimates at the high exposure levels are with low confidence due to the limited sample size (Figures 4 and S3). Therefore, future studies are needed to confirm or refute our findings on the health effect of large fires.” (L165-177).

19. Line 370 – Nothing in this paper is testing toxicity; I would avoid using this term.
[Response] Suggestion accepted. We removed all appearances of toxicity.
20. Line 371 – The estimates state here do not match those in the rest of the paper (1.4% compared to 5.1% in Table 1).
[Response] To avoid ambiguity, we modify the relevant sentences as “According to the adjusted model, a 1- $\mu\text{g}/\text{m}^3$ increment in the non-fire PM_{2.5} was associated with a 1.4% (1.1%, 1.6%) increase in pregnancy loss. The association was lower than that for fire PM_{2.5} (5.1%; 95% CI: 3.5 ~ 6.7%, Table 1).” (L137-139).
21. Line 374/375 – Where would I find the numbers to support this claim? Figure 5?
[Response] Yes. Accordingly, we add a notation at the end of the relevant sentence, which reads as “Additionally, in different countries, the importance of fire PM_{2.5} varied. The attributable fraction of fire PM_{2.5} was highest in Bangladesh, followed by India and Pakistan (Figure 5).” (L141-143).
22. Lines 411-413 – “Among various adverse birth outcomes, pregnancy loss may be the most dangerous because it not only reflects the most severe damage to the fetus but also will increase the risk of other outcomes in subsequent deliveries” If this is the case you should probably adjust the statistical models for whether the pregnancy loss occurred before or after the control.
[Response] No. To clarify that, we modify the sentence as “Among various adverse birth outcomes (*e.g.*, preterm birth, and low birthweight) that have been associated to PM_{2.5} exposure^{7, 8}, pregnancy loss may be the most dangerous because it not only reflects the most severe damage to the fetus but also will increase the risk of other outcomes in subsequent deliveries (*e.g.*, subsequent preterm birth)⁴².” (L188-191). Here, we aimed to explain why pregnancy loss should be at top of the priority list, when looking at different adverse outcomes of fire smoke exposure.
23. Line 417 – Again, you haven’t studied toxicity in the biological sense. Avoid using the term “toxicity” in reference to this work.
[Response] Suggestion accepted. We replace the word toxicity with “OR for per-unit exposure” (L194-195), accordingly.
24. Line 449-451 – The statement “The estimated association was not sensitive to differences in terms of covariate adjustment, subpopulation stratification, inclusion criteria for controls, and exposure assessment indicators” is simply not true. The CTM results are clearly sensitive to maternal age, and the results using other exposure metrics yield different answers depending on which covariates are used. Please be more careful in your wording.
[Response] We thank the reviewer for point out the imprecise statement. Accordingly, we

revise the conclusion as “In a subcontinental study in South Asia, we found a significant association between pregnancy loss and maternal exposure to fire-sourced PM_{2.5}. The association was not sensitive to differences in terms of covariate adjustment, inclusion criteria for controls, stratification by most of subpopulation factors (except for maternal age), but its significance level was changed by using alternative exposure indicators. Hence, we consider that the finding is moderately robust.” (L242-248).

Reviewer #2

[General comments] In “Open fire exposure increases the risk of pregnancy loss in South Asia,” the authors effectively describe findings from a study investigating associations between fires and pregnancy loss. This is an outstanding manuscript of high public health importance. The methods are clearly described, comprehensive, and innovative. The sensitivity analyses are well-justified and described. The results point to the critical importance of reducing exposure to air pollution during pregnancy. The following minor revisions are suggested.

[Response] The authors are encouraged a lot for the positive comments from the reviewer. Thank you.

[Specific comments]

1. Suggest that the authors use “participants” instead of “subjects” throughout the manuscript.
[Response] We replace all “subject(s)” with “participant(s)”, as suggested.

2. Minor typo is “risk” in first paragraph.

[Response] Thanks for point it out. We have corrected it in the revised manuscript.

3. The authors provide a link and relevant references to the DHS surveys in the first two paragraphs of the Methods. However, it would be helpful to briefly mention response rates and the sampling approach. For example, is the DHS able to reach all populations within the countries of interest. Are there particular populations that were excluded or difficult to reach?

[Response] Suggestion accepted. We add a new table (Table S2, as shown below) to document the detailed response rates and a few sentences. Now the revised sentences read as “The DHS surveys are household-based instruments, and the samples were selected using a complex two-stage design. For each national survey, in the first stage, enumeration areas are selected according to census data; in the second stage, households are sampled from an updated list of households. The females of reproductive age (15–49 years) in each household were of particular interest, and their records for socioeconomic status, fertility, reproductive history, infant mortality, etc. were screened by well-trained interviewers using standard questionnaires. For eligible female participates, the response rate varied by countries and ranged from 93-99% for the incorporated surveys (Table S2).” (L264-271). Since the response rate is high, it seems that the DHS doesn’t ignore a specific group of eligible women.

Table S2 Response rates for the surveys incorporated into this study.

Country	DHS phase*	Response rate of the eligible woman (%)		
		Urban	Rural	Total
India	7	95.8	97.0	96.7
Pakistan	5	93.3	95.3	94.5
	7	93.2	95.3	94.3
Bangladesh	4	98.3	98.8	98.6
	5	98.1	98.5	98.4
	6	97.0	98.4	97.9
	7	97.5	98.1	97.9

4. The spatial area or buffer is not clear for the exposure metrics derived from satellite remote sensing and fire emissions data. For example, within what area does the percent burned area apply?

[Response] Suggestion accepted. To illustrate that, we add a few words in the methods

section. Now the revised sentences read as “we assigned monthly series of open fire PM_{2.5} concentrations and non-fire PM_{2.5} concentrations to each participant, through spatially matching the residential address (longitude and latitude) geocoded by DHS with the corresponding pixel in the gridded PM_{2.5} maps. For comparative purpose, the alternative exposure indicators (*i.e.*, fire emissions and satellite images) were prepared in the same way as the PM_{2.5} preparation.” (L386-390). Here, we didn’t apply the buffer approach for satellite image or fire emission data. Applying multiple buffers to tune performance the exposure indicators is of interests but beyond the scope of our study, which relies on fire PM_{2.5} as the major exposure indicator.

5. The authors examine effect modification by parity in sensitivity analyses. Why not also examine parity as a potential confounder? Did the authors consider time between births as well?

[Response] We didn’t include, birth order, parity (which was derived from birth order) or the time from the first birth as a confounder, because those variables contained duplicated information, given the incorporation of maternal age and calendar year of delivery in the conditional regression model. For instance, between a case and its paired control, the difference in calendar year is equivalent to the time between the two pregnancies. Therefore, we refer not to include those variables as confounder to avoid over-adjustment. That also explains why we utilized the subset analysis rather than interaction analysis to explore the influence of parity. Interaction analysis usually requires the modifier also to be incorporated as a covariate (or confounder) or be matched up in a conditional model. For parity, such a requirement cannot be met. To illustrate that and to enhance the description on sensitivity analysis of parity, we add a few sentences, which read as “In the subset defined by parity, either cases or controls were required to be nulliparous or multiparous (*i.e.*, we excluded the case-control pairs which were not matched up in terms of parity). For every case-control pair in the nulliparous subset, the first successful delivery (*i.e.*, control) occurred following the corresponding pregnancy loss (*i.e.*, case).” (L441-445).

6. Figure 1. Please define abbreviations.

[Response] Relevant abbreviations (“PM_{2.5}: particulate matter with a diameter of less than 2.5 μm; DJF: December-January-February; MAM: March-April-May; JJA: June-July-August; SON: September-October-November.” [L598-599]) have been added.

7. First paragraph: Please describe how you defined “rural.”

[Response] Thank for the reminder. After a deep investigation, we find there is no uniform definition for urban-rural in DHS (“<https://userforum.dhsprogram.com/index.php?t=msg&th=55&start=0>”). To illustrate that, we add a footnote in Table S1, which describes the variables involved in this study. The footnote reads as “In DHS surveys, the definition of urban-rural is questionnaire-based and country-specific.”. Therefore, we realize the detailed interpretation based on this inconsistent definition of urban-rural may be inappropriate, and remove the relevant sentence in results section.

8. Save last sentence of “Associations by different exposure indicators” section for Discussion.

[Response] We remove the sentence to Discussion section (L217-218), as suggested.

9. Save last sentence of second to last paragraph of “Sensitivity analyses...” section for Discussion.

[Response] According to the suggestion, we delete the last sentence of the second paragraph, and remove most of last paragraph to the Discussion section (L165-177). We prefer to keep the core finding on nonlinear analysis (a few sentences in last paragraph of the section of “Sensitivity analyses for the estimated associations”) in the results section (L125-131). If the reviewer has more specific suggestion on organization of the paper, we

would like to do more modifications.

10. Please provide additional discussion of the following:

- a. What information might be lost by excluding women who experienced pregnancy loss but no live births?

[Response] Suggestion accepted. Excluding women who experienced pregnancy loss only may introduce bias, given a heterogeneous true effect of PM_{2.5}. In the discussion section, we add a few words to discuss that. The sentences read as “Finally, as this study was based on selected samples that were eligible for the self-compared design, representativeness of our results should be questioned. In a previous study³⁰, we use a statistical simulation to show that for a homogeneous true association, the self-compared design itself doesn’t introduce bias into its estimate by selecting samples. However, our sensitivity analysis suggests the association between fire PM_{2.5} and pregnancy loss may be heterogeneous (Figure 3). Therefore, selecting samples (e.g., excluding samples with pregnancy losses only might lead to ignoring the younger females or the individuals who were highly susceptible to the effect of fire PM_{2.5}) could bias the estimate away from the average effect, and thus limited representativeness of our findings.” (L229-236).

The relevant results from the simulation analyses that incorporated into our previous study are also documented below. The relevant results (highlighted by the red box) show that the self-compared design itself doesn’t introduce bias into its estimate by selecting samples.

Figure. Monte Carlo simulations (a) and relevant results (b). (a) Inputted parameters for the simulations are shown in italics. The estimates of the PL rate and spontaneous abortion rate were set according to a previous study. The relative error was calculated as $(\beta^* - \beta) \div \beta \times 100\%$, where β denotes the true value, and β^* denotes the corresponding estimation based on a simulated dataset. The results highlighted by red box shows how the estimator is affected by the self-compared design.

- b. Differences in toxicity of different types of open fires.

[Response] Suggestion accepted. We add a few words to discuss how the health effect varies with different types of fires. The relevant sentences read as “We also found a super-linear exposure–response between fire PM_{2.5} and pregnancy loss (Figure 4). The result suggests that the health impacts from large open fires, such as forest fires, should be considered. Our analysis of type-specific associations between fire emissions and pregnancy loss showed a consistent result (Figure S3). Stronger associations were found for the large fires (*i.e.*, grassland, shrubland fires, and temperate forest fires) compared to the small fires (*i.e.*, deforestation and degradation). However, the interpretation should be cautious for two reasons: first, by quantifying the exposure using fire PM_{2.5} concentrations, our analyses ignored the complex behavior of individual fires, which can be different in terms of size, duration and speed⁹. All dimensions of fire behavior can affect the exposure

pattern and chemical species of fire smoke, and thus the relevant health effect. Second, since frequency for large fires is low, our estimates at the high exposure levels are with low confidence due to the limited sample size (Figures 4 and S3). Therefore, future studies are needed to confirm or refute our findings on the health effect of large fires.” (L165-177).

- c. The importance of timing of exposure during pregnancy. For example, were women more sensitive during the third trimester?

[Response] Suggestion accepted. We add a few words on why this study has a limited capability to look at how the effect of PM_{2.5} varies at different trimesters. The relevant sentences read as “Third, the pregnancy losses were self-reported, so the reliability of the related data should be questioned due to a few reasons (e.g., recall bias). Although the reproductive history questionnaire has been validated in preliminary studies^{34, 35}, some outcomes could be still misclassified in the analyzed surveys. Also, uncertainty imbedded into the self-reported exposure time-window (*i.e.*, the gestational period) impede to explore how effect of PM_{2.5} varied between different trimesters. For instance, although the finding that miscarriage was more strongly associated with PM_{2.5} than stillbirth suggests more adverse for the exposure during early gestational stage, the estimates were with low confidence due to potentially misclassifying the two outcomes by recalling the gestational periods (Figure 3).” (L218-229).

11. In the last paragraph of the Discussion, please comment on the likely direction of the bias that could result from the limitations described.

[Response] Suggestion accepted. We add a few words to comment the potential direction of the bias. The relevant sentences read as “Given the above limitations, these results should be interpreted cautiously, and it is challenging to evaluate magnitude or direction of the total bias that is jointly caused by multiple limitations. Although our previous simulation analysis³⁰ suggested that a combination of some limitations might result in an underestimated association, further studies with advanced designs should be performed to evaluate our findings.” (L239-241). For the simulation analysis in our previous study, please also refer to the above Figure.

Reviewer #3

[General comments] The authors use health data and 3 indicators of PM_{2.5} associated with fires to determine health risk to pregnant women of exposure to PM_{2.5} from fires in South Asia. The results from this work and the focus region will be of general interest to a wide audience. I would recommend publication, after providing additional details about the methods used and results obtained. These are specified below.

[Response] The authors are encouraged a lot for the positive comments from the reviewer. Thank you.

[General comments]

1. Climate change is invoked throughout the manuscript as the cause for trends and variability in open fires in South Asia, but it is well known that severely degraded air quality in northern India is due to burning of agricultural waste (see for example: <https://doi.org/10.1038/s41598-019-52799-x>).

[Response] We thank the reviewer for the reminder. Accordingly, we modify all the relevant statements. The revised sentences read as “Interactions between climate change and anthropogenic activities result in increasing numbers of open fires, which have been shown to harm maternal health.” (L17-18), “Open fires comprise several different types including wildfires, mountain fires, coal mining fires, and slash-and-burn agriculture, and the sources can be directly related to human activities² or indirectly related via climate change³.” (L31-34), and “Driven by interactions between climate change and anthropogenic activities, frequent wildfires can dramatically affect many aspects of human sustainability⁴¹, including air quality, ecological diversity, distribution of infectious disease vectors, and public health.” (L178-180).

2. Is it surprising that the 3 metrics used give corroborative results, given that these datasets are not independent? GEOS-Chem uses the GFED inventory for open fire emissions, and GFED uses satellite-derived burned area to determine the location and extent of fires.

[Response] We agree with the reviewer, and add a few words to state the relationships between the three indicators. The relevant sentences now read as “Therefore, in this study, we used three different methods to evaluate maternal exposure to open fire smoke during gestation: satellite-based, emission-based, and CTM-based indicators of open fires, each of which has some advantages. Among the three, satellite-based indicators are the least influenced by artificial errors; emission-based measures incorporate small fires and can distinguish different fire types; and CTM-based indicators are the most easily interpreted, as they quantify exposure using PM_{2.5} concentrations, which makes the effect of open fires comparable to that of common air pollutants. It is worth to highlight that those three indicators are not independent of each other. Satellite measurements are incorporated into the GFED emissions together with other information (e.g., small fires due to agricultural waste burning), and the emissions are part of the inputs into the CTM. From satellite to emission, then to CTM, the complicated modelling procedures improve interpretability of the exposure indicator, but introduce potential artificial errors into the estimates as a cost to decouple the complexities.” (L186-298).

Since the three indicators have the same aim to quantify the fire exposure, it is not surprising to see the corroborative results, given that the underlying association between fire and pregnancy loss is physically true. However, we compared the three indicators, mainly in order to examine whether the estimated association was sensitive to the artificial errors (e.g., inaccurate quantification of small fires emissions, or inaccurate climate fields) embedded in the complex satellite-emission-CTM modelling procedure.

3. The description of the model includes superfluous information (“can be freely accessed from the website...” when referring to the MERRA-2 meteorology) and is missing relevant details. The latter includes the domain sampled (should be global for the coarse

resolution given), the time range that the model was simulated, the amount of time the model was spun up before sampling, and how the relevant primary and secondary aerosol components are represented in the model.

[Response] We thank the reviewer for the suggestion. Now we add more details about the GEOS-Chem model as follows: “In this study, we used the GEOS-Chem version 11-01 driven by assimilated meteorological fields from the NASA Global Modeling and Assimilation Office’s Modern-Era Retrospective analysis for Research and Applications Version 2 (MERRA-2). The gridded product has a spatial resolution of $0.5^\circ \times 0.625^\circ$ and can be freely accessed from the website of the NASA Goddard Earth Sciences Data and Information Services Center (<https://disc.gsfc.nasa.gov/>). The GEOS-Chem model has a spatial resolution of $2^\circ \times 2.5^\circ$ and 47 vertical layers. The model was run with the full O_x - NO_x - CO - VOC - HO_x chemistry, which includes sulfate-nitrate-ammonium, primary and secondary carbonaceous aerosols, mineral dusts and sea-salts. Specifically, sulfate-nitrate-ammonium is simulated through the ISOROPIA-II thermo-dynamical equilibrium²³. The aerosol simulations have been extensively evaluated using measurement data^{24, 25, 26}. The global anthropogenic emission inventory Community Emissions Data System (CEDS)¹⁷ was used to drive the GEOS-Chem model, and the fire emissions were taken from GFED4. We conducted GEOS-Chem simulations from 2000–2014 with a six-month spin-up starting from July 1999. $PM_{2.5}$ in the bottom layer were taken to represent the ambient $PM_{2.5}$ concentrations.” (L333-346).

4. The papers that the authors reference in support of model evaluation are from much earlier versions of the model and don’t necessarily evaluate the model over the study region.

[Response] Suggestion accepted. Beside the reference, we further evaluated the GEOS-Chem model using two approaches. To state that, we add a new paragraph, which reads as “The GEOS-Chem simulated $PM_{2.5}$ concentrations over our study domain were evaluated by (1) ground-surface monitoring data from five Indian cities (i.e., Chennai, Delhi, Hyderabad, Kolkata, and Mumbai), and (2) the satellite-based $PM_{2.5}$ estimates. The monitoring data were obtained from the U.S. Embassy and Consulates in India (<https://in.usembassy.gov/embassy-consulates/new-delhi/air-quality-data/>), and were aggregated as the monthly averages during 2013-2014. The GEOS-Chem simulations were in good agreement with the in-situ observations ($R^2 = 0.69$, Figure S4), the gold-standard values for exposure assessment. Since the spatiotemporal coverage of monitoring data was limited, we further compared the GEOS-Chem simulations with the satellite-based $PM_{2.5}$ estimates²⁷, which were obtained from a global annual product during 2000–2014 with a spatial resolution of $0.1^\circ \times 0.1^\circ$. The satellite-based $PM_{2.5}$ estimates²⁷, which have been reported to be in good correlation ($R^2 = 0.81$) with ground-surface concentrations, were moderately correlated with annual averages of our simulations (R^2 was varied by year from 0.47 to 0.60, Table S3).” (L347-359).

5. In the description of the model calibration: What satellite-based product do the authors use? Include an appropriate reference for this. At what spatial resolution is the calibration conducted? Does the calibration typically correct for a low or high bias in the model? What is the size of this bias?

[Response] Suggestion accepted. We a few words to enhance the relevant statements. The revised sentences now read as “Additionally, considering the uncertainties in emission inventories and modeling procedures (such as parametrization of chemical reactions, and model boundary conditions)²⁸, bias correction with ground-surface observations has been widely applied to improve the performance of CTM in exposure assessment²⁹. Therefore, we introduced the annual mean of satellite-based $PM_{2.5}$ estimates²⁷ ($PM_{2.5,y}^{satellite}$), which have used in our previous studies^{10, 30} exploring the association between total $PM_{2.5}$ and pregnancy loss, as a constraint to calibrate our GEOS-Chem outputs.” (L366-371) and “We downscaled the GEOS-Chem outputs ($PM_{2.5,m,y}^{wfire}$, $PM_{2.5,m,y}^{nofire}$ and $\rho_{m,y}$) to the $0.1^\circ \times 0.1^\circ$ grid using an inverse-distance-weighting approach, to match them with the satellite-based $PM_{2.5}$ estimates. After the calibration, the performance of the GEOS-Chem

simulations ($PM_{2.5, m, y}^{wfire}$) was improved, according to all statistical metrics of the comparison with monitoring data (Figure S1). For instance, the mean bias was reduced from 26.6 to 20.8 $\mu\text{g}/\text{m}^3$ after the calibration.” (L380-385).

6. Why does a small fire $PM_{2.5}$ contribution of 2% result in a large impact on health (13% of disease burden associated with $PM_{2.5}$)?

[Response] According to the question, to enhance the relevant explanations, we add a few words. Now the revised sentences read as “We found that the excess risk of fire $PM_{2.5}$ was 269% (149%, 411%) higher than that of non-fire $PM_{2.5}$ (Figure 5). Because of the high OR for per-unit exposure¹³, open fires contributed only a small amount to the concentration of $PM_{2.5}$ (2%) but a non-negligible fraction of the $PM_{2.5}$ -associated disease burden (13%).” (L193-196). For a brief illustration, if the disease burden can be quantified as $\text{Disease} = \beta_{\text{fire}} \times \text{Exposure}_{\text{fire}} + \beta_{\text{non-fire}} \times \text{Exposure}_{\text{non-fire}}$, and then fraction for exposure (x) and its disease burden (y) can be derived as $x = \text{Exposure}_{\text{fire}} / (\text{Exposure}_{\text{fire}} + \text{Exposure}_{\text{non-fire}})$ and $y = (\beta_{\text{fire}} \times \text{Exposure}_{\text{fire}}) / (\beta_{\text{fire}} \times \text{Exposure}_{\text{fire}} + \beta_{\text{non-fire}} \times \text{Exposure}_{\text{non-fire}})$, respectively. x is estimated as 2%, while y is estimated as 13%. Next, we can get the following equation:

$$\frac{1}{y} = \frac{1}{x} + \left(\frac{\beta_{\text{non-fire}}}{\beta_{\text{fire}}} - 1 \right) \times \frac{\text{Exposure}_{\text{non-fire}}}{\text{Exposure}_{\text{fire}}}.$$

When $\frac{\beta_{\text{non-fire}}}{\beta_{\text{fire}}} < 1$, we can get $\frac{1}{y} < \frac{1}{x}$ or $x < y$. The brief mathematical calculation can help to explain why a small fire $PM_{2.5}$ contribution of 2% result in a large impact on health.

[Specific comments]

7. Abstract: consider replacing “pre-established database” with “inventory” and “referent” with “reference”.

[Response] We would like to modify those words as suggested. However, in the revised manuscript, those sentences are removed in order to keep the abstract within the words limit (< 150 words).

8. Methods description of Population data: In the first paragraph “CTM-based open fires $PM_{2.5}$ concentrations were available” is provided as the reason for focusing on 2000 to 2014, but the model isn’t limited to these years.

[Response] To clarify that, we modify the relevant sentence as “Individual records on women’s reproductive history were collected from all available demographic and health surveys (DHS) in India, Pakistan, and Bangladesh from 2000 to 2014, when multiple exposure indicators were simultaneously available (The anthropogenic inventories¹⁷ inputted into our CTM are until 2014, and the satellite remote sensing measurements of fire points are from 2000, as described below.)” (L252-256).

9. In the paragraph starting “Table 1 presents the associations...”, what does “model adjustments” mean? In the last sentence of this paragraph, could the very coarse spatial resolution of the model be a contributing factor?

[Response] We revise a few sentences relevant to the reviewer’s questions. The model adjustment means the covariates in regression model. To clarify that, we modify the relevant sentence as “However, the significance level of the ORs estimated by satellite data on burned area or dry-matter emissions data was sensitive to adjustments of different covariates.” (L101-102). We agree with the reviewer that the coarse spatial resolution is also a possible explanation. To state that, we modify the relevant sentence as “A possible explanation is the potential for misclassification of exposure (due to multiple reasons, e.g., spatial resolution or accuracy) when using satellite or emissions data.” (L102-104). According to the theory of epidemiology, the coarse spatial resolution is another reason to cause exposure misclassification (e.g., the exposure hotspot is falsely classified as a lower level by averaging the surrounding values in the scenario of low spatial resolution.).

10. In Fig. 2, what does “P = ” represent?

[Response] According to the question, at the end of the caption, we add a sentence, which reads as “In each panel, ‘P’ presents the likelihood ratio test P-value for an alternative hypothesis that the estimated associations are different between subgroups classified by a subpopulation indicator.” (L603-605).

11. In the Discussion, what is “common PM_{2.5}”?

[Response] To clarify that, we replace the term by “urban PM_{2.5} pollution” (L183 and L186).

Reviewers' Comments:

Reviewer #1:

Remarks to the Author:

Thank you for addressing my previous comments. The work was already good but is even better now. I have two minor comments to add but am otherwise satisfied:

- 1) The description of the three exposure methods seems extraneous since almost all of the main health model results were estimated using the CTM exposure metric. It should be made clearer that the CTM metric is the main metric.
- 2) Line 426: I don't understand why you are controlling for calendar year under this study design.

Reviewer #2:

Remarks to the Author:

The authors have addressed fully all of my comments and concerns. I have no further suggestions for revision. This is an important paper.

Reviewer #3:

Remarks to the Author:

I am overall satisfied that the authors have done a careful job of addressing my review comments and requests for clarity and additional information.

My only remaining minor specific comments include:

- (1) Change "ISORROPIA-II thermo-dynamical equilibrium" to "ISORROPIA-II thermodynamic equilibrium model". Note that ISORROPIA has 2 R's.
- (2) Change "R2 was varied by year from 0.47 to 0.60" to "R2 varied each year from 0.47 to 0.60".

To reviewers,

We thank the reviewers for the 2nd round evaluations on our manuscript. The responses are listed point-by-point as follows. The revisions corresponding to specific comments are highlighted by blue color in our responses below.

Reviewer #1

[Specific comments]

1. The description of the three exposure methods seems extraneous since almost all of the main health model results were estimated using the CTM exposure metric. It should be made clearer that the CTM metric is the main metric.

[Response] Suggestion accepted. To clarify that, we add a new sentence in introduction section and another in methods section. The former reads as “**Due to good interpretability, the CTM-based indicator was used as the main metric in our epidemiological analyses.**”; and the latter reads as “**Since interpretability is critical to assess the health impacts from open fires, we utilize the CTM-based indicator as main metric in our epidemiological analyses.**”.

2. Line 426: I don't understand why you are controlling for calendar year under this study design.

[Response] In global scale, the maternal health was improved year by year due to many reasons, including economic growth, development of new technology and etc.. Those unmeasured factors together resulted in a long-term trend in the health outcome (Lawn et al. 2016). The calendar year (a pair of case and control could happen at different years) could be a good indicator for the trend. The similar approaches (e.g., dummy variable or smoothed spline term of calendar year) have been utilized in many environmental epidemiological studies (e.g., Shi et al. 2016; Neidell 2004).

To enhance the relevant statement, we modified the relevant sentences as “**We used a set of calendar-year-specific intercepts to further control for the long-term trend (e.g., improvement of maternal health driven by economic development) in the outcome.**”.

Reference

Lawn JE, Blencowe H, Waiswa P, Amouzou A, Mathers C, Hogan D, Flenady V, Frøen JF, Qureshi ZU, Calderwood C, Shiekh S. Stillbirths: rates, risk factors, and acceleration towards 2030. *The Lancet*. 2016 Feb 6;387(10018):587-603.

Shi L, Zanobetti A, Kloog I, Coull BA, Koutrakis P, Melly SJ, Schwartz JD. Low-concentration PM2.5 and mortality: estimating acute and chronic effects in a population-based study. *Environmental health perspectives*. 2016 Jan;124(1):46-52.

Neidell MJ. Air pollution, health, and socio-economic status: the effect of outdoor air quality on childhood asthma. *Journal of health economics*. 2004 Nov 1;23(6):1209-36.

Reviewer #3

1. Change "ISORROPIA-II thermo-dynamical equilibrium" to "ISORROPIA-II thermodynamic equilibrium model". Note that ISORROPIA has 2 R's.

[Response] Thanks for pointing out the typo. We correct it in the revised version.

2. Change "R² was varied by year from 0.47 to 0.60" to "R² varied each year from 0.47 to 0.60".

[Response] Thanks for pointing out the typo. We correct it in the revised version.